# Rapid and simultaneous detection of *Campylobacter* spp. and *Salmonella* spp. in chicken samples by duplex loop-mediated isothermal amplification coupled with a lateral flow biosensor assay

**Thanawat Sridapan**[1], **Wanida Tangkawsakul**[2], **Tavan Janvilisri**[3],
**Taradon Luangtongkum**[4], **Wansika Kiatpathomchai**[5], **Surang Chankhamhaengdecha**[6]*

1 Graduate Program in Molecular Medicine, Faculty of Science, Mahidol University, Bangkok, Thailand,
2 Center of Nanoscience and Nanotechnology, Faculty of Science, Mahidol University, Bangkok, Thailand,
3 Department of Biochemistry, Faculty of Science, Mahidol University, Bangkok, Thailand, 4 Department of Veterinary Public Health, Faculty of Veterinary Science, Chulalongkorn University, Bangkok, Thailand,
5 Bioengineering and Sensing Technology Research Team, National Center for Genetic Engineering and Biotechnology (BIOTEC), National Science and Technology Development Agency (NSTDA), Pathum Thani, Thailand, 6 Department of Biology, Faculty of Science, Mahidol University, Bangkok, Thailand

* surang.cha@mahidol.ac.th

## Abstract

Development of a simple, rapid and specific assay for the simultaneous detection of *Campylobacter* spp. and *Salmonella* spp. based on duplex loop-mediated isothermal amplification (d-LAMP), combined with lateral-flow biosensor (LFB) is reported herein. LAMP amplicons of both pathogens were simultaneously amplified and specifically differentiated by LFB. The specificity of the d-LAMP-LFB was evaluated using a set of 68 target and 12 non-target strains, showing 100% inclusivity and exclusivity. The assay can simultaneously detect *Campylobacter* and *Salmonella* strains as low as 1 ng and 100 pg genomic DNA per reaction, respectively. The lowest inoculated detection limits for *Campylobacter* and *Salmonella* species in artificially contaminated chicken meat samples were $10^3$ CFU and 1 CFU per 25 grams, respectively, after enrichment for 24 h. Furthermore, compared to culture-based methods using field chicken meat samples, the sensitivity, specificity and accuracy of d-LAMP- LFB were 95.6% (95% CI, 78.0%-99.8%), 71.4% (95% CI, 29.0%-96.3%) and 90.0% (95% CI, 73.4%-97.8%), respectively. The developed d-LAMP-LFB assay herein shows great potentials for the simultaneous detection of the *Campylobacter* and *Salmonella* spp. and poses a promising alternative approach for detection of both pathogens with applications in food products.

## Introduction

Foodborne pathogens can cause serious adverse effects via contaminated food or water. Particularly, foodborne bacteria, *Campylobacter* and *Salmonella* are recognized as major causative

**Data Availability Statement:** All relevant data are within the paper and its S1–S4 Figs, S1 Raw images, S1–S4 Tables files.

**Funding:** This study was supported by the International Research Network (IRN) under The Thailand Research Fund (TRF) [grant numbers IRN59W0007]. TS received a research assistantship scholarship from the Faculty of Graduate Studies, Mahidol University, Academic Year 2018.

**Competing interests:** The authors have declared that no competing interests exist.

agents of human foodborne enteritis and death worldwide [1]. The number of cases of *Campylobacter* and non-typhoidal *Salmonella* was estimated at >95 and >78 million foodborne illnesses worldwide, respectively [1]. Both pathogens are highly prevalent in poultry, especially commercial chicken meat, which is often implicated as the main food vehicle of infection for human through the consumption of raw or undercooked contaminated poultry meat and products [2–5]. Transmission of these organisms from poultry to humans is a serious public health threat. To prevent outbreaks of foodborne illness, microbiological testing is necessary to monitor food products in order to yield satisfactory quality according to the standard procedures and regulatory guidelines [6–8], assuring the safety and quality of food production for human consumption.

Culture-based methods are the gold standard for detecting these pathogens present in food, however it is time and labor consuming [9]. Numerous PCR-based approaches are most widely used in laboratories with an improvement by reducing the time required to obtain the results. In particular, a large number of commercially available real-time PCR kits have been developed as an implementation for rapid diagnostic technique of foodborne pathogens [10]. However, most detection kits are able to detect only a single pathogen either *Campylobacter* or *Salmonella* species. The time and reagent cost for independently detecting these pathogens were increased, required sophisticated instrument, and highly trained personnel to carry out the test, rendering it difficult to be implemented in the resource-limiting areas. Hence, rapid, cost-effective and simultaneous detection tools for *Campylobacter* and *Salmonella* in food samples would be valuable for food industry and regulatory agencies.

Loop-mediated isothermal amplification (LAMP) has been widely used to overcome the drawback of those assays because it is performed under constant temperature with high sensitivity, specificity and rapidity for the low-cost detection of pathogens [11, 12]. To this date, an advancement of LAMP method, namely multiplex LAMP (m-LAMP) has been increasingly applied for simultaneous detection of multiple target genes in a one-tube reaction. The limitation of this approach is to distinguish m-LAMP amplicons by gel electrophoresis as a result of mixed ladder patterns. Therefore, the results of the assay require confirmation by coupled with various analytical techniques [12] to specifically distinguish them including restriction enzyme digestion and gel electrophoresis [13], melting temperature analysis [14], fluorescent dye-conjugated agents [15] and lateral-flow biosensor (LFB), which was a favoured technique to apply for end-point detection of LAMP products [16–21]. LFB is a simple and rapid detection method that does not require advanced instruments [18]. Thus, m-LAMP combined with a LFB would be useful as a considerable time and cost-saving assay for simultaneous detection of multiple pathogens. To date, there has been no report on the use of the d-LAMP-LFB for the simultaneous detection of *Campylobacter* and *Salmonella*.

Thus, in this study, d-LAMP assay was developed for simultaneous detection of *Campylobacter* and *Salmonella* spp. The product complexes of LAMP amplicons were clearly analyzed using LFB. The d-LAMP-LFB assay also was evaluated the assay performance in field chicken meat samples comparing to culture-based method. The schematic illustration of the use of d-LAMP-LFB assay is depicted in Fig 1.

## Materials and methods

### Bacterial strains

A total of 80 strains of bacteria were used in this study (S1 Table). *C. jejuni* DMST 15190 and *Salmonella enterica* serovar Typhimurium ATCC 23566 were used as standard strains to develop d-LAMP-LFB assay. *Campylobacter* spp. was grown in sheep blood agar (Clinical Diagnostic, Thailand) at 37°C for 48 h in a microaerophilic condition (8% $O_2$, 7% $CO_2$

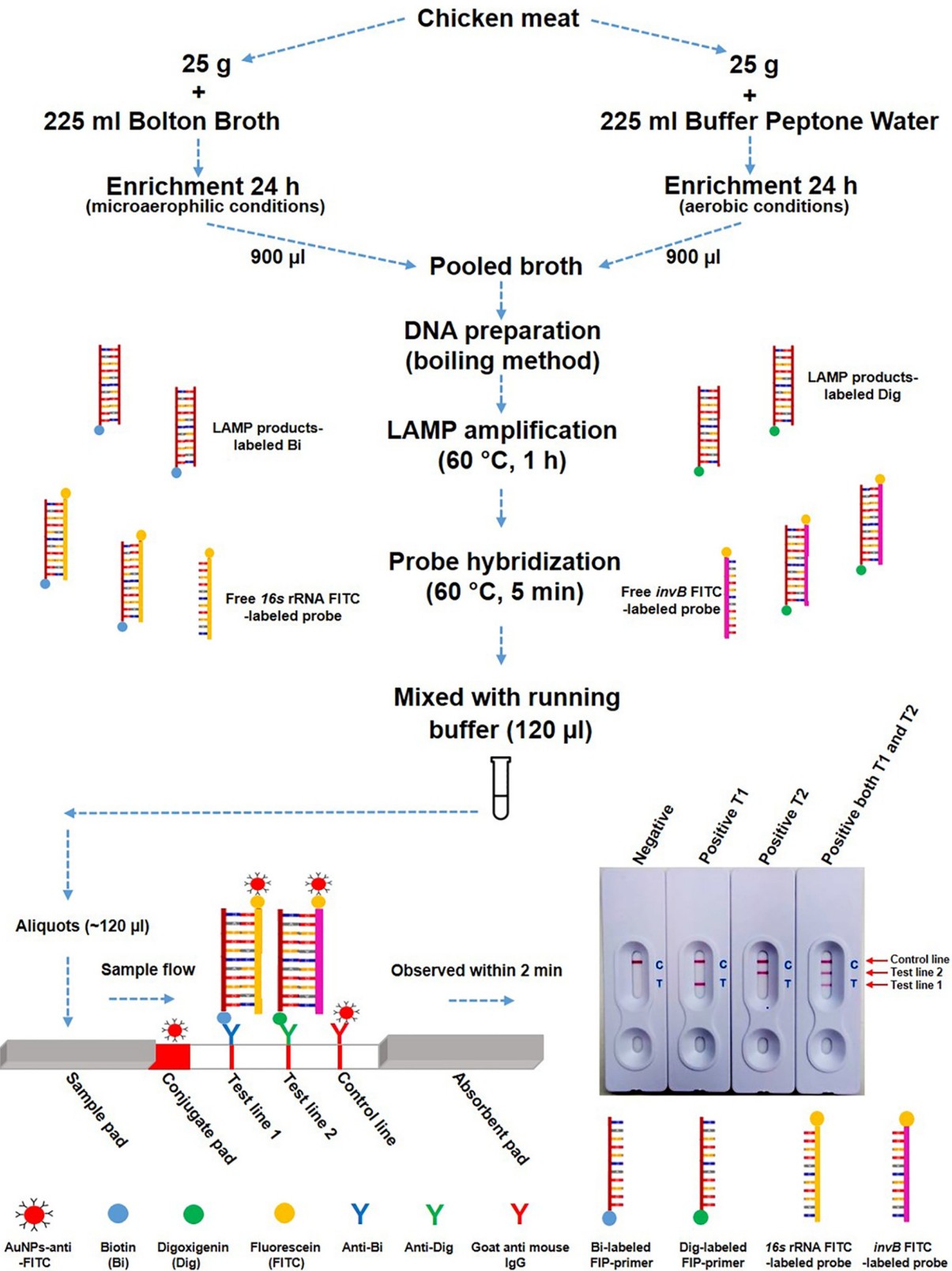

**Fig 1. Schematic depiction of d-LAMP-LFB for simultaneous detection of *Campylobacter* and *Salmonella* spp.**

and 85% $N_2$) created with the Anaero Pack®-MicroAero (Mitsubishi Gas Chemical Co., Inc., japan), and at 37°C overnight for *Salmonella* spp. and other bacterial strains. For DNA preparation, a single colony of each strain was suspended in 5 ml Brain Heart Infusion (BHI) broth (Himedia, India) and then microaerophilically incubated at 37°C overnight for *Campylobacter* species, and aerophilic condition with agitation for other bacterial strains.

## DNA preparation

Cultures were centrifuged at 15,500 *g* for 5 min at room temperature (RT). The supernatant was discarded and the cell pellet was resuspended with 100 μl of 1×Tris EDTA (TE) buffer, pH 8.0, boiled at 100°C for 15 min to release the bacterial DNA, and then immediately chilled on ice for 5 min. The mixture was centrifuged at 18,000 *g* for 10 min at RT. The supernatant was used as DNA template for the LAMP assay [21]. DNA concentration was quantified using the NanoVue Plus TM spectrophotometer (Biochrom., USA). In order to assess the analysis of analytical sensitivity, genomic DNA of *C. jejuni* DMST 15190 and *S.* Typhimurium ATCC 23566 were ten-fold serially diluted from 50 ng down to 0.5 fg for single species detection as well as the mixture of both DNA templates were prepared by mixing with equal concentration, 10-fold continuously diluted from 50 ng to 0.5 fg and subjected to d-LAMP-LFB for detection of two targets in a reaction.

## LAMP primer and probe design

*Campylobacter* spp. 16S ribosomal RNA (16S rRNA) (GenBank accession no. AL111168.1) and *Salmonella* spp. invasion protein B (*invB*) (GenBank accession no. CP009102.1) encoding DNA regions were used as targets. 16S rRNA and *invB* genes were determined as highly conserved sequences within *Campylobacter* and *Salmonella* strains, respectively [22–24]. The nucleotide sequences were aligned using Multiple Sequence Comparison by Log-Expectation (MUSCLE) program to select the conserved regions for all either *Campylobacter* or *Salmonella* species. Based upon the conserved regions presented in those alignments, a set of four LAMP primers including F3, B3, FIP and BIP was designed using Primer Explorer V5 software (http://primerexplorer.jp/lampv5e/index.html), then tested with BLAST at the NCBI database (http://blast.ncbi.nlm.nih.gov/Blast.cgi) and selected to provide 100% specificity. Additionally, LAMP primers of 16S rRNA were checked *in silico* using BioEdit software (http://www.mbio.ncsu.edu/bioedit/bioedit.html) to discriminate closely related genera including *Arcobacter* and *Helicobacter*, from *Campylobacter* species. A poly T linker "TTTT" was added in the FIP and BIP between F1c - F2 and B1c - B2. FIP-16S rRNA was labeled with biotin, while FIP-*invB* was labeled with digoxigenin at 5' end. The detection probe labeled with fluorescein iso-thiocyanate (FITC) at 5' end was designed to anneal to the central region between the F1c and B1c primer targets of the LAMP amplicons (Table 1). All primers and probes were synthesized by Bio Basic Inc., Canada and reconstituted in sterile distilled water to 100 μM stock solution.

## Determination of optimal LAMP-LFB conditions

The reaction of single-plex LAMP (s-LAMP) assay was performed in 25 μl of a mixture containing 1×ThermoPol® Reaction Buffer (20 mM Tris–HCl, 10 mM KCl, 2 mM $MgSO_4$, 10 mM $(NH_4)_2SO_4$, 0.1% Triton X-100), 6 mM $MgSO_4$, 1.4 mM of dNTP mix (Vivantis, Malaysia), 0.2 μM each of F3 and B3 primer, 1.6 μM each of FIP and BIP primer, 8 U of *Bst* DNA polymerase, large fragment (New England Biolabs Inc., USA). 2 μl (20 ng) DNA template of *C. jejuni* DMST 15190 and *S.* Typhimurium ATCC 23566 were used as positive controls. A

**Table 1. Nucleotide sequences of d-LAMP-LFB primers and probes used in this study.**

| Primer | Length | Sequence (5'-3') |
|---|---|---|
| F3-16S rRNA | 20 | CGATCTGCTGGAACTCAACT |
| B3-16S rRNA | 18 | CATGCTCCACCGCTTGTG |
| [a]FIP-16S rRNA | 42 | TAGGGCGTGGACTACCAGGG-TTTT-GACGCTAAGGCGCGAAAG |
| BIP-16S rRNA | 46 | ACGCATTAAGTGTACCGCCTGG-TTTT-GGTCCCCGTCTATTCCTTTG |
| [c]Probe-16S rRNA | 20 | CTAGTTGTTGGGGTGCTAGT |
| F3-*invB* | 20 | CGGAAAAGAAGCGACAGAGG |
| B3-*invB* | 19 | CGGGCGACATTTGACAGAT |
| [b]FIP-*invB* | 43 | GGCTTTCGCTTTGGCGTCTG-TTTT-GCCTTAGATAAGGCCACGG |
| BIP-*invB* | 44 | TGACCAAATTCCAGGGAACGGC-TTTT-CTGCTCACCCTGGGAAAC |
| [c]Probe-*invB* | 19 | GAGAAAGCGGATAACATTC |

[a] 5'-modified with biotin.

[b] 5'-modified with digoxigenin.

[c] 5'-modified with FITC.

reaction with 2 μl sterile distilled water was performed as the blank control. The reaction was incubated at 60˚C for 1 h and then terminated the amplification at 85˚C for 5 min.

For d-LAMP-LFB assay, the optimized reaction was employed in 25 μl as described above. The constant temperature of 58–65˚C and final primer concentration for each target were adjusted. Then, LAMP products were hybridized with FITC-labeled DNA probe (0.2 μM each of 16S rRNA and *invB*), following the optimal temperature delineated above for 5 min and then inactivated the reaction at 85˚C for 5 min.

For visualization of LAMP results [21], 0.5 μl of hybridized LAMP products were added into microcentrifuge tube containing 120 μl of running buffer (PBS and Surfynol® 465 surfactant) (Kestrel Bio Sciences Thailand Co. Ltd). All mixture volume was added onto the sample pad of LFB, which was fabricated and prepared by Kestrel Bio Sciences Thailand Co. Ltd. The solution migrated along the LFB strip through the conjugate pad, which was coated with gold nanoparticles labeled with anti-FITC. The complexes could then be captured by the anti-biotin embedded on the test line 1, anti-digoxigenin on the test line 2 and goat anti mouse IgG on the control line. After 2 min, the results were visualized by observing bands that appear on the T1, T2 and/or C line. The positive result was defined when either T1 or T2 and C lines were observed. A single visible on the C line was interpreted as a negative result. Meanwhile, the result was invalid when the C line did not appear. In addition, the obtained hybridized LAMP products were subjected on a 2% agarose gel electrophoresis preparing with 1×Tris-Borate EDTA buffer, stained with Serva DNA stain G (SERVA Electrophoresis GmbH, Germany). The gel image was visualized under UV light using the mini UV table ultraviolet analyzer (Extragene, Taiwan) to evaluate the difference ladder-like banding pattern.

## Specificity of d-LAMP-LFB assay

The specificity of the d-LAMP-LFB assay was evaluated under the optimal condition using 20 ng each of the isolated DNA templates from 80 bacterial strains described earlier, including *Campylobacter* spp. (n = 33), *Salmonella* spp. (n = 35) and other bacterial strains (n = 12) (S1 Table). The test was performed at least twice and included a reaction with 2 μl sterile distilled water as a blank control.

## Determination of the lowest inoculated detection limit in artificially contaminated samples

To prepare the bacterial suspensions for spiking food samples, the initial cell concentration was adjusted as the turbidity of an overnight culture to 0.5 McFarland (Grant Bio™ Densitometer, UK) together with spread plating count on sheep blood agar, which corresponded to $10^8$ and $10^7$ colony forming units (CFU)/mL for *C. jejuni* DMST 15190 and *S.* Typhimurium ATCC 23566, respectively. The dilution series (McFarland 0.5 down to $10^{-8}$) of each strain were prepared in 0.85% saline solution, to obtain suspensions with the number between 1 to $10^8$ CFU/mL of *C. jejuni* and 0.1 to $10^7$ CFU/mL of *S.* Typhimurium.

To demonstrate the applicability of d-LAMP-LFB assay, the food samples were prepared for spiking study as described previously [25] with some modifications. Chicken meat samples were obtained from a supermarket and surface rinsed with sterile water for irrigation (A.N.B. Laboratories Co., LTD, Thailand). Two portions of chicken samples were aseptically cut and weighed to twenty-five grams, followed by surface wash with sterile water for irrigation, dip with 70% ethanol solution for 30 s, rinsed with water for irrigation and finally evaporated under UV light for 30 min. The prepared food matrices were spiked with 100 μl of the desired level of either *S.* Typhimurium or *C. jejuni* placed in a sterile stomacher bag (BagPage Plus 400, Interscience, France) containing 225 mL of Buffer Peptone Water (BPW, Himedia, India) for *Salmonella* spp. spiked study, and 225 mL of Bolton broth (BB, CM0983) supplemented with Modified Bolton Broth Selective Supplement (SR0208E, Oxoid, USA, cefoperazone 10 mg/500 mL, vancomycin 10 mg/500 mL, trimethoprim 10 mg/500 mL and amphotericin B 5 mg/500 mL) for *Campylobacter* spp. spiked study. The mixture was homogenized for 1 min at low speed (BagMixer 400 Lab Blender, Interscience, France). Food sample with 100 μl of 0.85% NaCl was included as negative assay control for the absence of either *Campylobacter* or *Salmonella* species in the prepared food materials. The incubation time required for enrichment culture step was performed according to the ISO 6579–1:2017 [26] and ISO 10272–1:2017 [27] with some slight modifications. *S.* Typhimurium-spiked cultures were incubated at 37°C for 24 h without shaking whereas *C. jejuni*-spiked cultures were incubated under microaerophillic conditions at 37°C for 4 h and then for an additional 20 h at 42°C.

DNA preparation were performed as previously described [21, 28, 29] with a few modifications as follows. 900 μl each of dilution of enrichment culture of *C. jejuni* and *S.* Typhimurium was pipetted and pooled in a 2 ml of microcentrifuge tube and then centrifuged at 90 *g* for 3 min at RT to remove larger food matrices. The supernatant (1,500 μl) was transferred to a new tube and centrifuged at 15,500 *g* for 5 min at RT to precipitate bacterial cells. The supernatant was discarded and 500 μl of 1× phosphate buffer saline (PBS), pH 7.2 was added to resuspend and vortexed. Then, the mixture was centrifuged at 15,500 *g* for 5 min at RT, and the supernatant was discarded. The pellet was resuspended with 100 μl of 1 TE buffer, boiled for 15 min, and immediately chilled on ice for 5 min. Finally, the mixture was centrifuged at 18,000 *g* for 10 min at RT. The supernatant containing DNA template was collected and stored at -20°C until use. The analytical sensitivity test was performed in three replicates, and the last dilution in each spiked sample that test positive was considered as the lowest inoculated detection limit [25].

## Analytical sensitivity of the d-LAMP-LFB assay

In order to determine the analytical sensitivity of s- and d-LAMP-LFB, both serially diluted genomic DNA templates from pure culture and serially enriched broth-extracted genomic DNA of *C. jejuni* and *S.* Typhimurium from spiked food samples were prepared as described earlier for confirming the analytical sensitivity. The genomic DNA amount of the templates including single and multiple targets were subjected to LAMP-LFB assay as aforementioned.

## Evaluation of the d-LAMP-LFB assay using naturally contaminated food samples

To evaluate the efficacy of d-LAMP-LFB assay, raw chicken meat samples were obtained from the retail markets (n = 30) in Bangkok, Thailand (S4 Table). All chicken samples were kept at 4˚C and transported immediately to the laboratory. The sample was aseptically removed from the package, weighted as twenty-five grams and transferred in a sterile stomacher bag containing 225 mL of BPW for enrichment of *Salmonella* spp. and 225 mL of BB supplemented with SR0208E for enrichment of *Campylobacter* spp. Each bag was homogenized at low speed for 1 min and then incubated for 24 h as aforementioned. For the d-LAMP-LFB assay of the enriched sample, 900 μl of both enriched media were pooled for DNA extraction as describe before. The results of d-LAMP-LFB assay were also compared to culture-based method for *Salmonella* spp. and *Campylobacter* spp. by the ISO 6579–1:2017 [26] and ISO 10272–1:2017 [27], respectively.

For isolation and identification of *Salmonella* spp., 0.1 mL and 1 mL of enriched BPW were inoculated into the 10 mL of Rappaport-Vassiliadis broth (RV) and Tetrathionate broth (TT) (Himedia, India), respectively. Then, the RV was incubated at 42˚C, while TT at 37˚C for 24 h. Then, 10 μl of RV and TT were streaked onto Xylose Lysine Deoxycholate (XLD) agar (Himedia, India) and incubated at 37˚C for 24 h. A portion of three suspected colonies were examined and resuspended in a microcentrifuge tube containing 100 μl of 1×TE buffer, then DNA extraction was performed by using boiling method as describe earlier. Identification of *Salmonella* spp. was performed by detecting a 262 bp DNA fragment of the *invA* gene specific for *Salmonella* spp. [30, 31] (S2 Table).

For isolation and identification of *Campylobacter* spp., 10 μl of BB was streaked onto Bolton agar supplemented with SR0208E and incubated at 42˚C for 48 h under microaerophillic conditions. The maximum of three suspicious colonies were selected and extracted genomic DNA. Confirmation of the colonies as *Campylobacter* spp. was performed by amplification of a 1,062 bp of 16S rRNA [30, 32] (S2 Table).

## Statistical analysis

Results from d-LAMP-LFB were compared to the detection performance to those of standard culture-based method. The sensitivity, specificity and accuracy were calculated with 95% confidence intervals (95% CIs) using MedCalc (https://www.medcalc.org/calc/diagnostic_test.php).

# Results

## Optimization of d-LAMP-LFB

To determine the optimal condition for d-LAMP-LFB assay, two parameters such as isothermal temperature and final primer concentration were optimized individually and visualized by LFB. The results showed that an optimal multiplex temperature of 60˚C gave a clear visible red band for T1, T2, and C lines. At the same time, the amount of primers was adjusted for simultaneous amplification of many targets in a single reaction. To give the optimal results, the final primer concentration for each target of d-LAMP was similar to s-LAMP-LFB assay, allowing equal amplification of all primer sets. The optimal primer concentrations were shown in the following: 0.2 μM each of 16S rRNA and *invB* F3 and B3 primer, 1.6 μM each of 16S rRNA and *invB* FIP and BIP primer. Thus, these optimized reactions were employed in the d-LAMP-LFB assay throughout this study.

## Detection and differentiation of multiple LAMP products

Under the optimized conditions described above, each LAMP product from s-LAMP of either *S.* Typhimurium or *C. jejuni* was subjected to agarose gel electrophoresis. Two different typical pattern-like bands were observed and distinguished by the lowest position of ladder formation in relation to the DNA marker (192 bp of *S.* Typhimurium and 220 bp of *C. jejuni*) (Fig 2A). Comparatively, d-LAMP generated multiple LAMP amplicons of both pathogens, which could not be distinguished by agarose-gel patterns. To overcome the limitation, the developed LFB successfully differentiated a mixture of LAMP amplicons, which was hybridized with designed FITC labelled DNA probes. In addition, the detection probe recognized only their targets of specific LAMP amplicons with no cross-reactivity with each other (Fig 2B). These results demonstrated that LAMP in combination with LFB was effective for visually simultaneous detection of both pathogens.

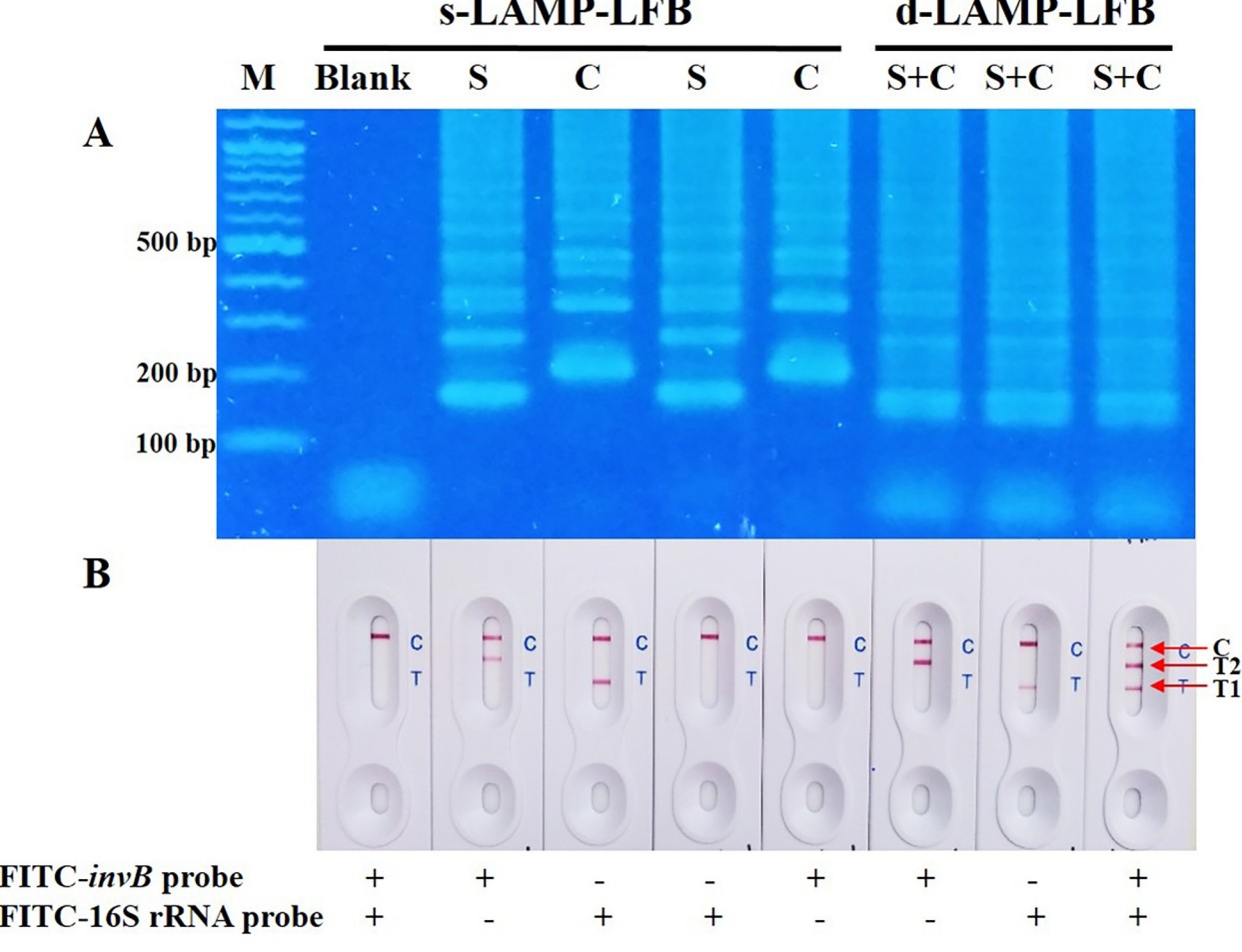

**Fig 2. Detection and differentiation of LAMP products visualized by 2% agarose gel electrophoresis in comparison to LFB.** (A) Electrophoretic analysis represented the distinctive ladder-like patterns of LAMP amplification products for S: *S.* Typhimurium; C: *C. jejuni*; SC: *S.* Typhimurium + *C. jejuni*; M: 2-log DNA ladder 100 bp; Blank: the reaction with 2 μl sterile distilled water. (B) LFB analysis showed the simultaneous detection of multiple targets with distinguishing by visualizing band that appear on the C: control line; T1: test line 1 of *C. jejuni*; T2: test line 2 of *S.* Typhimurium.

## Specificity of d-LAMP- LFB assay

To determine the specificity of two LAMP primer sets. The developed d-LAMP-LFB assay was applied to detect all the test strains of these two species including 33 *Campylobacter* spp., 35 *Salmonella* spp. and 12 other bacterial strains. The positive results were correctly identified only when genomic DNA of *Campylobacter* or *Salmonella* spp. was used in a reaction. No visualize signal on the test line was examined in any other bacterial strains and blank control (Figs 3 and S1 and S1 Table). This newly developed two sets of primers was able to specifically amplify 16S rRNA and *invB* gene of the *Campylobacter* spp. and *Salmonella* spp., respectively. Therefore, these results demonstrated that d-LAMP-LFB exhibited 100% inclusivity and exclusivity and was reliable for simultaneous detection of *Campylobacter* spp. and *Salmonella* spp.

## Determination of analytical sensitivity

To evaluate the analytical sensitivity for a single target by s-LAMP-LFB assay, the sensitivity limit of both pathogens was 100 pg per reaction (Fig 4A and 4B). Meanwhile, to simultaneously detect multiple targets by d-LAMP-LFB assay, the sensitivity limit of *C. jejuni* and *S.* Typhimurium were 1 ng and 100 pg DNA per reaction, respectively (Fig 4C). Notably, a less sensitive as 10-fold in sensitivity limit for *C. jejuni* was observed when dilutions containing DNA mixtures were amplified simultaneously compared with the separate amplification, while there were no significant differences in efficiency between single- and multiplex analyses of *S.* Typhimurium.

The theoretical analytical sensitivity of the assay was further verified in spiked raw chicken samples, *Campylobacter* and *Salmonella* negative raw chicken meat samples were spiked with either *C. jejuni* or *S.* Typhimurium, respectively. For s-LAMP-LFB assay, the lowest inoculated

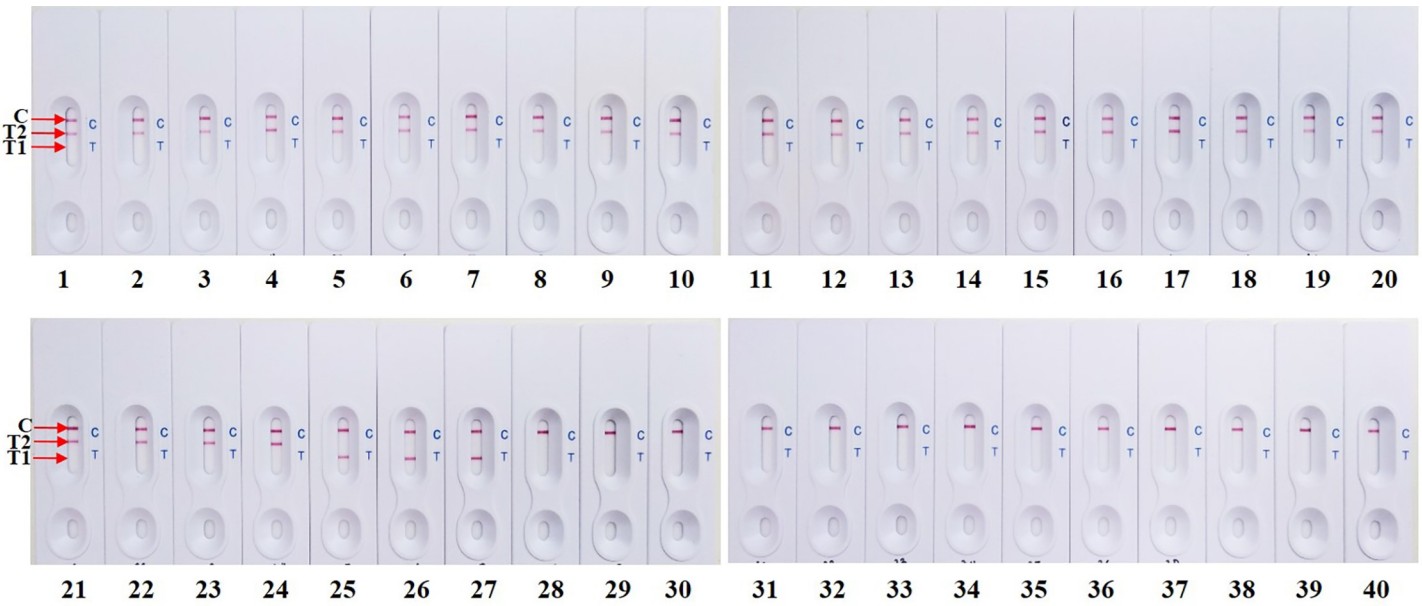

**Fig 3. The specificity of d-LAMP-LFB assay for detection of different strains of 20 ng each of DNA templates.** 1: *S.* Agona DMST 10638; 2: *S.* Abony DMST 21863; 3: *S.* Anatum DMST 16870; 4: *S.* Arizonae DMST 22439; 5: *S.* Bangkok DMST 7121; 6: *S.* Bergen DMST 10895; 7: *S.* Cerro DMST 17381; 8: *S.* Derby DMST 8535; 9: *S.* Enteritidis DMST 15676; 10: *S.* Gallinarum DMST 15968; 11: *S.* Hvittingfoss DMST 15681; 12: *S.* Mbandaka DMST 17377; 13: *S.* Newport DMST 15675; 14: *S.* Panama DMST 10640; 15: *S.* Paratyphi B DMST 28118; 16: *S.* Poona DMST 15679; 17: *S.* Schwarzengrund DMST 17364; 18: *S.* Senftenberg DMST 17013; 19: *S.* Stanley DMST 16874; 20: *S.* Typhi DMST 5784; 21: *S.* Typhimurium ATCC 23566; 22: *S.* Typhimurium DMST 562; 23: *S.* Wandsworth DMST 19204; 24: *S.* Waycross DMST 19205; 25: *C. jejuni* DMST 15190; 26: *C. coli* DMST 18034; 27: *C. lari* DMST 17953; 28: *B. cereus* ATCC 14579; 29: *E. aerogenes* DMST 2720; 30: *E. coli* DMST 703; 31: *E. coli* DMST 4212; 32: *L. monocytogenes* DMST 17303; 33: *S. boydii* DMST 30245; 34: *S. aureus* ATCC 25923; 35: *S. epidermidis* DMST 15505; 36: *S. haemolyticus* DMST 15511; 37: *V. cholera* DMST 2873; 38: *V. vulnificus* DMST 21245; 39: *Y. enterocolitica* DMST 8012; 40: blank control.

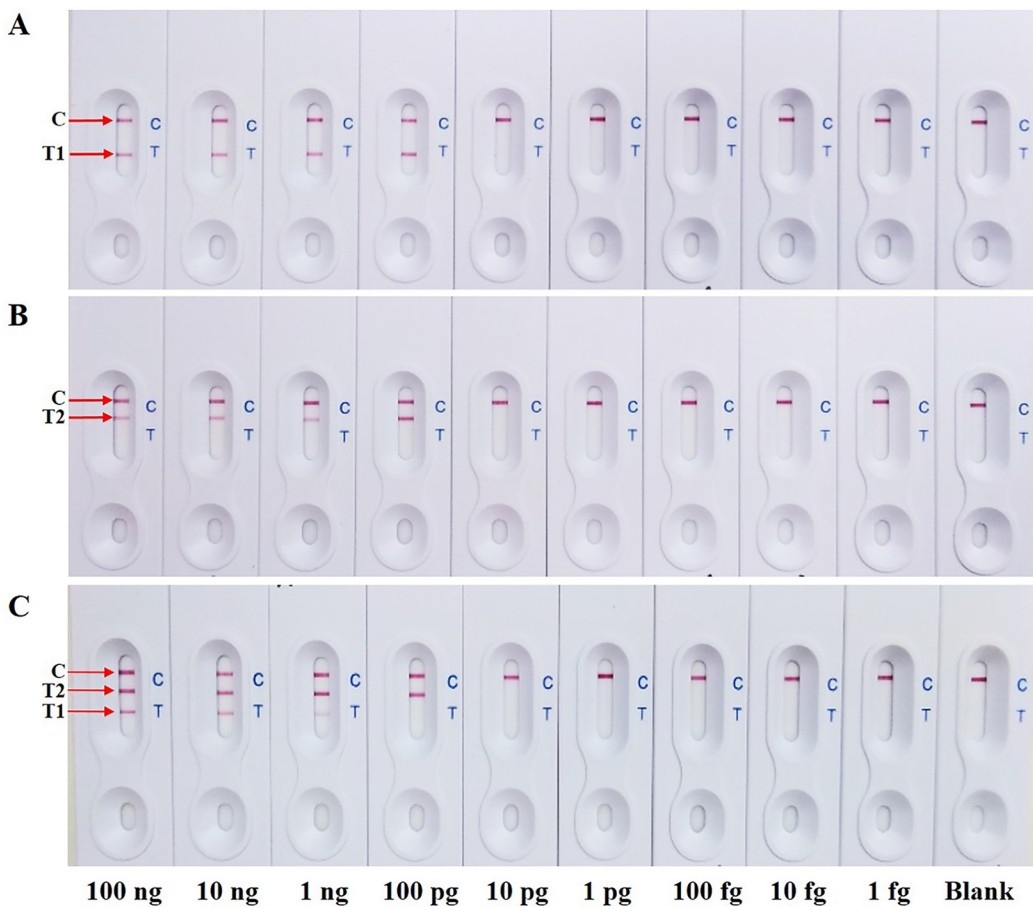

**Fig 4.** Analytical sensitivity of s-LAMP-LFB assay for a single target detection of (A) *C. jejuni* or (B) *S.* Typhimurium using each of serial dilution of target DNA. (C) Sensitivity limit of d-LAMP-LFB assay applied for simultaneous detection of both pathogens using 10-fold serial dilution of mixed genomic DNA ranged from 100 ng -1 fg per reaction. C: control line; T1: test line 1 of *C. jejuni*; T2: test line 2 of *S.* Typhimurium.

detection limits of *C. jejuni* and *S.* Typhimurium were $10^2$ (Fig 5A) and 1 CFU per 25 g (Fig 5B), respectively. Comparatively, the lowest inoculated detection limits of d-LAMP-LFB assay for simultaneous detection of *C. jejuni* and *S.* Typhimurium were $10^3$ and 1 CFU per 25 g, respectively (Fig 5C). These results indicated that the lowest inoculated detection limit of d-LAMP-LFB assay for detecting *S.* Typhimurium was consistent with s-LAMP-LFB assay approach, while lowest inoculated detection limit of *C. jejuni* was 10-fold less sensitive than that of a single target detection.

## Evaluation of the d-LAMP-LFB assay

To evaluate the feasibility of our d-LAMP-LFB assay, a total of 30 raw breast chicken meat samples collected from retail markets were used to evaluate the d-LAMP-LFB assay by comparing to those obtained using culture based-method. Among all 30 samples, 22 samples were consistently detected positive by both assays. Of these 22 d-LAMP-LFB-positive samples, 1 sample was positive for *Salmonella* spp. only, 10 samples were positive for *Campylobacter* spp. only, and 11 samples were contaminated with both *Campylobacter* and *Salmonella*. Furthermore, 2 samples were identified as false-positive d-LAMP-LFB results for *Salmonella* spp., 1 sample was observed as false-negative detection for *Campylobacter* spp. The remaining 5

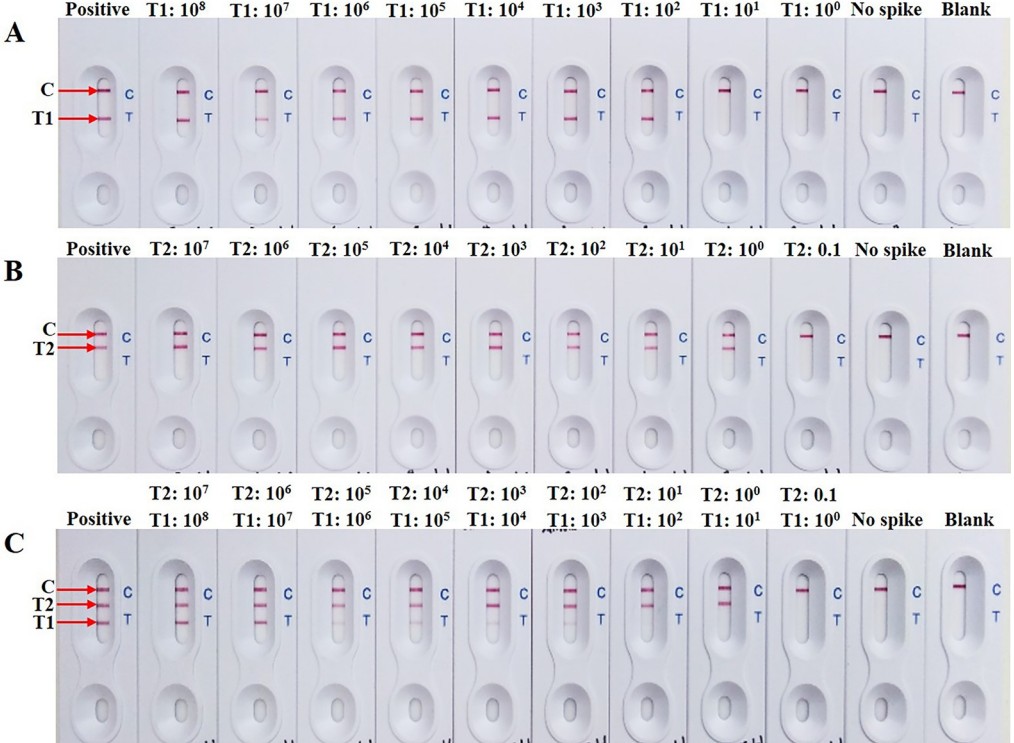

**Fig 5.** Lowest inoculated detection limit of s-LAMP-LFB assay for a single target detection of (A) *C. jejuni* or (B) *S.* Typhimurium in artificially contaminated raw chicken meat samples after 24 h enrichment. (C) Lowest inoculated detection limits of d-LAMP-LFB assay for simultaneous detection of both pathogens. Positive: positive control; No spike: non-spiked bacterial; Blank: blank control; C: control line; T1: test line 1 of *C. jejuni*; T2: test line 2 of *S.* Typhimurium.

samples were *Campylobacter*/*Salmonella*-negative by both d-LAMP-LFB and culture method. Compared to the standard culture results, the sensitivity, specificity and accuracy for the d-LAMP-LFB assay were 95.6% (95% CI, 78.0%-99.8%), 71.4% (95% CI, 29.0%-96.3%) and 90.0% (95% CI, 73.4%-97.8%), respectively. An evaluation assay of all 30 samples is displayed in Figs 6 and S2–S4 and S3 and S4 Tables.

## Discussion

We reported herein the development of a simple and rapid assay for simultaneous detection of *Campylobacter* and *Salmonella* species in foods based on the d-LAMP-LFB. Successful amplification relies on the specificities of two sets of LAMP primers and probes, which specifically detected 80 bacterial strains. For the lowest inoculated detection limit in spiking study, it was claimed that $10^3$ CFU *Campylobacter* and 1 CFU *Salmonella* per 25 g of sample could be simultaneously detected within 2 h, time taken after 24 h of enrichment culture. In addition, this assay can be applied to examine food samples contaminating these pathogens quickly, conveniently and reliable alternative for culturing.

As the cause of most of the acute gastroenteritis in humans worldwide, *Campylobacter* and *Salmonella* are identified as major causative pathogens [1]. For Campylobacteriosis, the thermotolerant species *C. jejuni* and *C. coli* are the most frequently reported in the number of case of human infections [33]. Our result showed that developed d-LAMP-LFB assay specifically detected three thermophilic *Campylobacter* species (*C. jejuni*/*C. coli*/*C. lari*) (Fig 3 and S1 Table) as well as in all

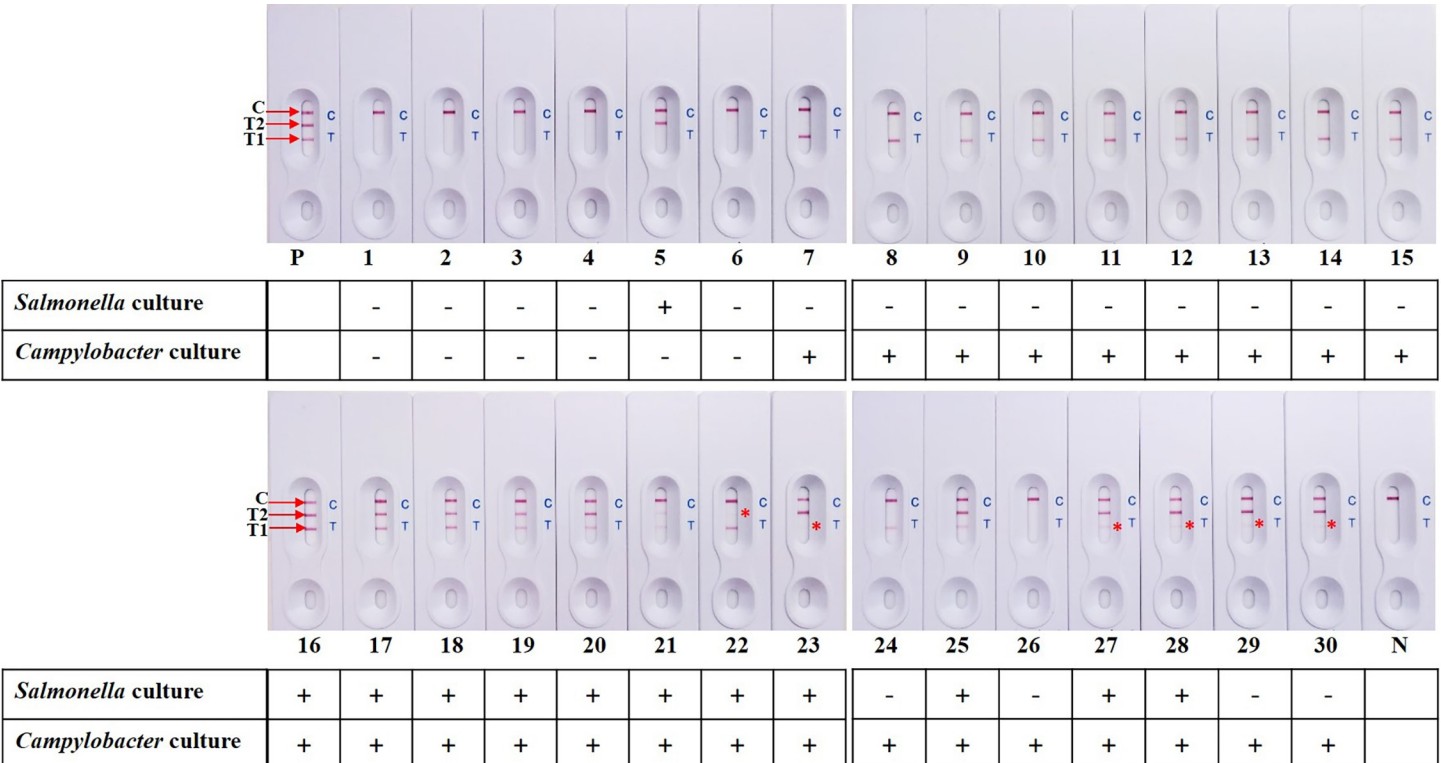

| | P | 1 | 2 | 3 | 4 | 5 | 6 | 7 | 8 | 9 | 10 | 11 | 12 | 13 | 14 | 15 |
|---|---|---|---|---|---|---|---|---|---|---|----|----|----|----|----|----|
| *Salmonella* culture | | - | - | - | - | + | - | - | - | - | - | - | - | - | - | - |
| *Campylobacter* culture | | - | - | - | - | - | - | + | + | + | + | + | + | + | + | + |

| | 16 | 17 | 18 | 19 | 20 | 21 | 22 | 23 | 24 | 25 | 26 | 27 | 28 | 29 | 30 | N |
|---|----|----|----|----|----|----|----|----|----|----|----|----|----|----|----|---|
| *Salmonella* culture | + | + | + | + | + | + | + | + | - | + | - | + | + | - | - | |
| *Campylobacter* culture | + | + | + | + | + | + | + | + | + | + | + | + | + | + | + | |

**Fig 6. Evaluation of d-LAMP-LFB by detecting of *Salmonella* spp. and *Campylobacter* spp. in naturally contaminated raw chicken meat samples.** LFB: 1–30 obtained from retail markets; LFB 26 showed false-negative result for *Campylobacter* spp.; LFB 29 and 30 showed false-positive result for *Salmonella* spp.; P: positive control (*C. jejuni* DMST 15190 + *S.* Typhimurium ATCC 23566); N: blank control; * Indicated weakly positive signals; "+": the culture test result is positive; "-": the culture test result is negative.

30 *Campylobacter* isolates strains (S1 Fig and S1 Table), indicating that the newly developed d-LAMP-LFB assay was specific for all *Campylobacter* organisms. In addition, all *Salmonella* strains in six serogroups (A, B, C1, C2, D, and E) (S1 Fig and S1 Table) that cause approximately 99% of Salmonellosis in humans and warm-blooded animals [34] were successfully detected by d-LAMP-LFB assay. Within these serogroups, five *Salmonella* serovars, including *S.* Enteritids (group D), *S.* Typhimurium (group B), *S.* Infantis (group C1), *S.* Virchow (group C1) and *S.* Hadar (group C2), which are responsible in animals destined for human consumption [35] and prioritized by the European Commission (EU) for an examination and control of poultry and poultry products entry [36] were also sufficiently detected by our d-LAMP-LFB assay. Thus, developed d-LAMP-LFB assay in this study is potential use for the detection of all of both *Campylobacter* and *Salmonella* strains responsible for foodborne infections.

Many previous reports published only a single LFB assay for the detection of either *Campylobacter* or *Salmonella* spp. in food [37–39], as it is challenging to enrich both microorganisms simultaneously, due to the fact that their growth requirements on different selective broths as well as slower growth of *Campylobacter* under microaerobic conditions [40]. However, as previously described [41], both organisms contaminated in food were enriched independently and were then gathered for DNA extraction. In this current study, results obtained using the combined enriched step and d-LAMP-LFB assay successfully represented the simultaneous detection of both pathogens.

The major consideration to multiplex reactions is that the analytical sensitivity may be decreased with the combination of all primer sets within a single reaction. By observations in

both pure genomic DNA and spiking experiments, s- and d-LAMP-LFB results showed that there was no obvious difference in analytical sensitivity for *Salmonella* spp., whereas *Campylobacter* species was reduced by ten-fold in multiplex assay. It is possible that the amplification efficiency of the designed primers for *Campylobacter* strains might be lower than that of *Salmonella* spp. from primer dimers or secondary structures between the multiple primer sets in d-LAMP. It was therefore necessary to optimize the d-LAMP-LFB condition by adding some chemicals such as dimethyl sulfoxide (DMSO), which could elevate amplification efficiencies by inhibiting secondary structures in the DNA primers [13, 42]. However, the analytical sensitivity of our d-LAMP-LFB assay in spiking study was also comparable to the LFB in other previous studies. It has been reported that the lowest inoculated detection limit for *Salmonella* targeting *invA* gene was $8 \times 10^3$ CFU/25 g (320 CFU/g) after enrichment for 18–24 h [37], 36 CFU/25 g (1.44 CFU/g) targeting *hilA* gene after 6 of enrichment [38], while *C. jejuni* was $10^1$–$10^2$ CFU/ 25 g (or $10^1$–$10^2$ CFU/mL of initial inoculum levels) targeting *hipO* gene without enrichment step [39]. Several qPCR kits such as BAX® *Campylobacter coli/jejuni/lari* assay (DuPont Qualicon), BAX® *Salmonella* assay (Hygiena), BIOTECON foodproof® *Campylobacter* Detection Kit (Biotecon Diagnostics) and BIOTECON foodproof® *Salmonella* Detection Kit (Biotecon Diagnostics) are commercially available with the level of detection as low as 1–10 CFU/25 g sample or $10^3$–$10^4$ CFU/mL after 20–24 h and 24–48 h of enrichment for *Salmonella* and *Campylobacter*, respectively. Our assay showed the lowest inoculated detection limits as low as 1 CFU/25 g (0.04 CFU/g) and $10^3$ CFU/25 g (or $10^3$ CFU/mL of initial inoculum levels) for detecting of *Salmonella* and *Campylobacter*, respectively, which was in the same range to those of previous reports and commercialized kits. Even though a minimum enrichment time in this study required more time of 24 h, our d-LAMP-LFB assay offers a valuable detection tool for detecting two targets in a single tube, with a reduction in assay turnaround times, leaving some potential for complicated instrumentation and the experience of laboratory personnel.

To assess the feasibility of using the d-LAMP-LFB assay to detect *Campylobacter* and *Salmonella* contamination in field chicken meat samples, we detected 12 positives for *Salmonella* spp., 23 positives for *Campylobacter* spp. and 5 negatives for both. Comparing to the culture method, our d-LAMP-LFB assay resulted in two false-positives for *Salmonella* spp. Due to its high sensitivity, LAMP assay is highly susceptible to carryover contamination of the previously amplified products that lead to false positive results [12, 43]. Those samples were re-tested in a proper laboratory procedure at strictly-controlled set up areas and equipment for carryover contamination prevention. However, both samples were still tested positive. It is possible that d-LAMP-LFB assay might (i) detect dead cells found in the chicken samples, or (ii) detect viable but non-culturable [9] in XLD agar used in this study. According to standardized microbiological procedures [26, 44], *Salmonella* spp. can be cultured in other selective agar of choice like Hektoen Enteric agar (HE) or Brilliance *Salmonella* agar (BE). False-negative d-LAMP-LFB result for *Campylobacter* spp. was observed in one sample. This discrepancy was unclear whether it is truly false-negative results as the determination of the amplified LAMP product via LFB visual assessment using the naked-eye [16, 21]. The subjective interpretation of result could explain the limited detection sensitivity of LFB [45], especially when low numbers of *Campylobacter* spp. were present in food, which could not be detected by our assay with its lowest inoculated detection limit of $10^3$ CFU/ 25. To improve the sensitivity of d-LAMP-LFB assay, using another enriched medium like Preston enrichment broth, which allowed for a shorter enrichment period at 18–24 h may increase the sensitivity limit of *Campylobacter* [28]. Further studies are required to confirm this approach. Thus, an interpretation of test results must be carefully considered.

Our study showed that this d-LAMP-LFB assay is able to detect both pathogens present in chicken samples with enrichment culturing and poses as an alternative for standard culture methods. Notably, the results obtained from the culture method showed that raw chicken meat

samples from retail markets with pooled, unwrapped and stored together at ambient temperatures were contaminated with both *Campylobacter* and *Salmonella*. This can be explained by the fact that this type of packaging is susceptible to cross-contamination from an environment during the meat preparation process through other raw food, utensils and tools by food workers [46], whereas the retail markets that supplied by chilled/frozen-chicken manufacturers with good packaging with plastic wrap and stored by individual on a refrigerated shelf are more hygienic conditions, demonstrated that food storage facilities leading less source of potential contamination [47, 48]. Thus, the finding in this study suggested the need for improvement of hygienic practices in retail markets with poor packaging to ensure food safety and reduce the risk of foodborne pathogen infection from both pathogens.

In conclusion, we successfully developed the d-LAMP-LFB assay to simultaneously detect *Campylobacter* and *Salmonella* in one assay. This assay is sensitive, specific, accurate and cost-effective in comparison to the gold standard. d-LAMP-LFB assay could be used for rapid screening for *Campylobacter* and *Salmonella* spp. contamination in chicken meat and other food products at manufacture production lines. Thus, in this study, we first report the use of a d-LAMP-LFB assay for the detection of bacterial pathogens directly from food samples. However, further optimization and verification are required to evaluate the detection performance characteristics of this d-LAMP-LFB assay.

## Supporting information

**S1 Fig. The specificity of d-LAMP-LFB assay for detecting different strains of *Campylobacter and Salmonella* spp. using 20 ng each of DNA templates.** C and T indicate the control and test lines, respectively. A positive result displayed bands at both C and T, while a negative result showed one band at the C line. 1: *S.* Enteritidis DMST 17368; 2: *S.* Enteritidis DMST 33954; 3: *S.* Typhi DMST 22842; 4: *S.* Typhimurium DMST 2069; 5: *S.* Typhimurium DMST 16150; 6: *S.* Typhimurium DMST 16152; 7–36: *Campylobacter* spp. isolates (M-23, 26, 45, 51, 62, 91, 93, 97, 152, 165, 198, 213, 272, 351, 352, 363, 370, 379, 392, 399, 406, 413, 417, 430, 441, 445, 449, 450, 451 and 464); 37: *S.* Choleraesuis; 38: *S.* Hadar; 39: *S.* Infantis; 40: *S.* Paratyphi A; 41: *S.* Virchow.
(PDF)

**S2 Fig. Agarose gel electrophoresis of PCR products obtained from the DNA extracted from colonies in RV medium based-XLD agar for the detecting *Salmonella* spp. by culture based-method.** A: UFUL and URUL primer; B: Sal1598 F and Sal1859 R primer; Sample 1–30 obtained from the retail markets; Lane M: 2-log DNA ladder; Lane P: positive control (*S.* Typhimurium ATCC 23566); Lane N: blank control (the reaction with 2 μl sterile distilled water).
(PDF)

**S3 Fig. Agarose gel electrophoresis of PCR products obtained from the DNA extracted from colonies in TT medium based- XLD agar for the detecting *Salmonella* spp. by culture based-method.** A: UFUL and URUL primer; B: Sal1598 and Sal1859 R primer; Sample 1–30 obtained from the retail markets; Lane P: positive control (*S.* Typhimurium ATCC 23566); Lane N: blank control (the reaction with 2 μl sterile distilled water).
(PDF)

**S4 Fig. Agarose gel electrophoresis of PCR products obtained from the DNA extracted from colonies in Bolton agar for the detecting *Campylobacter* spp. by culture based-method.** A: UFUL and URUL primer; B: 16S-F and 16S-R primer; Sample 7–30 obtained from the retail markets; Lane P: positive control (*C. jejuni* DMST 15190); Lane N: blank control (the

reaction with 2 μl sterile distilled water).
(PDF)

**S1 Table. Bacterial strains used in this study to determine the specificity of the designed primers for d-LAMP-LFB assay.**
(PDF)

**S2 Table. Primer sequences and conditions used for the PCR assay.**
(PDF)

**S3 Table. Comparison of d-LAMP-LFB assay results and culture-based method for simultaneous detection of *Campylobacter* and *Salmonella* spp. in raw chicken meat samples.**
(PDF)

**S4 Table. The source of field chicken samples and results of the culture method for detection of *Campylobacter* spp. and *Salmonella* spp.**
(XLSX)

**S1 Raw images. Original photographs of Figs 2A and S2, S3 and S4.**
(PDF)

## Acknowledgments

The authors thank Dr. Watanalai Panbangred and Dr. Soraya Chaturongakul for *Salmonella* strains.

## Author Contributions

**Conceptualization:** Thanawat Sridapan, Surang Chankhamhaengdecha.

**Data curation:** Thanawat Sridapan.

**Formal analysis:** Thanawat Sridapan.

**Funding acquisition:** Wansika Kiatpathomchai, Surang Chankhamhaengdecha.

**Investigation:** Tavan Janvilisri, Surang Chankhamhaengdecha.

**Methodology:** Thanawat Sridapan, Wanida Tangkawsakul.

**Project administration:** Surang Chankhamhaengdecha.

**Resources:** Taradon Luangtongkum, Surang Chankhamhaengdecha.

**Supervision:** Surang Chankhamhaengdecha.

**Validation:** Thanawat Sridapan.

**Writing – original draft:** Thanawat Sridapan.

**Writing – review & editing:** Tavan Janvilisri, Surang Chankhamhaengdecha.

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
