## [Decision Letter · Decision Letter 0]

12 Apr 2021

PONE-D-21-06858

Rapid and simultaneous detection of Campylobacter spp. and Salmonella spp. in chicken samples by duplex loop-mediated isothermal amplification coupled with a lateral flow biosensor assay

PLOS ONE

Dear Dr. Chankhamhaengdecha,

Thank you for submitting your manuscript to PLOS ONE. After careful consideration, we feel that it has merit but does not fully meet PLOS ONE’s publication criteria as it currently stands.

Though reviewer 1 provided favorable comments, reviewer 2 raised some concerns especially the rationale for target genes, the probe design, assay specificity and time to result and recommended against its publication. However, given the importance of a need for novel rapid methods for foodborne pathogens, the current method utilizes both LAMP and lateral flow platforms together and provides an opportunity for an improved detection platform. Therefore, we are willing to reconsider our decision, provided you are willing to address all concerns raised by the reviewers.

We look forward to receiving your revised manuscript.

Kind regards,

Arun K. Bhunia, Ph.D.

Academic Editor

PLOS ONE

Additional Editor Comments:

Though reviewer 1 provided favorable comments, reviewer 2 raised some concerns especially the rationale for target genes, the probe design, assay specificity and time to result and recommended against its publication. However, given the importance of a need for novel rapid methods for foodborne pathogens, the current method utilizes both LAMP and lateral flow platforms together and provides an opportunity for an improved detection platform. Therefore, I am willing to reconsider my decision, provided the authors are willing to address all concerns raised by the reviewers.

Journal Requirements:

Reviewers' comments:

Reviewer's Responses to Questions

**Comments to the Author**

1. Is the manuscript technically sound, and do the data support the conclusions?

Reviewer #1: Yes

Reviewer #2: Partly

2. Has the statistical analysis been performed appropriately and rigorously? 

Reviewer #1: Yes

Reviewer #2: Yes

3. Have the authors made all data underlying the findings in their manuscript fully available?

Reviewer #1: Yes

Reviewer #2: Yes

4. Is the manuscript presented in an intelligible fashion and written in standard English?

Reviewer #1: Yes

Reviewer #2: Yes

5. Review Comments to the Author

Reviewer #1: The manuscript described a d-LAMP-LFB assay for rapid, sensitive and simultaneous detection of Campylobacter spp. and Salmonella spp.in the chicken samples in one assay. In my opinion, the assay is technically sound, and the experimental data support the conclusion. There is only one minor question needed to empharized.

In line 66 page 3, you mentioned that the real-time PCR kits for food-borne pathogens are costly. According to my knowledge, the real-time PCR is less expensive than the LAMP, especially the LAMP combined with lateral flow biosensor.

Reviewer #2: Comments for Manuscript Number PONE-D-21-06858

1. The authors indicated that the d-LAMP is rapid and simple for simultaneous detection of Salmonella spp. and Campylobacter spp. However, this technique must rely on pre-enrichment which is the bottle neck of the over all process and make it not rapid and simple as claimed by the authors. Though the pre-enrichment has the major advantage in discrimination between dead and alive pathogenic bacteria, but it is tedious and time- consuming. Hence, the overall process starting from sample treatment could take approximately 24 hours or more which is not rapid and simple. As such, line 398-391 should be deleted or re-discussed since the d-LAMP combined with pre-enrichment could not be applied for field investigation.

2. One of the major food-borne pathogenic bacteria is Listeria spp. which is often found in food products. Hence, d-LAMP of Salmonella spp. and Campylobacter spp. may not be sufficient for surveillance test.

3. The d-LAMP condition is arbitrary for detection of both Salmonella spp. and Campylobacter spp.as well as the selected gene for primer design. Hence, this may affect the sensitivity or limit of detection of each bacterial test. Generally, the LOD of d-LAMP is less sensitive than a single target detection.

Are there any reasons that the authors select 16 SRNA and invB for primer and probe design? What is/are the feature (s) or advantage (s) of these genes over the others?

4. The author should indicate the melting temperature of each primer in Table 1.

5. Lines 415-419 are repeated to each other. This should be re-written. In my opinion detection without pre-enrichment is much more challenge.

6. Lines 422-425, the authors should point out that the less LOD of Campylobacter spp. is the limitation of d-LAMP when compared to the single detection. Hence, the primers should be re-designed or changed the gene target,

7. Lines 445-451, the discussion of two false positive samples are conflict to the purposes of this study. Since the samples were pre-enriched prior to d-LAMP, the contamination of dead cells is not quite possible. The second reason as the author discussed that d-LAMP-LFD could detect viable but non-culturable in XLD agar indicated that the specificity of the test is questionable. So, this is another the limitation of the test.

8. Lines 459-463, The authors mentioned the improvement of sensitivity of Campylobacter spp. using d-LAMP-LFD assay by extending incubation time or change culture medium. This confirmed that the detection of Campylobacter spp. was not rapid and simple as indicated by the authors.

9. Lines 480-485, The discussion is unacceptable due to the d-LAMP LFD assay has some limitations as followed.

a. The assay required pre-enrichment step which is time-consuming and tedious.

b. The LOD of the assay is less sensitive than a single detection especially for Campylobacter spp.

c. The specificity of the assay is questionable since the viable non-culturable in XLD agar could interfere the results.

10. In conclusion of my comment, this confirmed that the d-LAMP-LFD for detection of Salmonella spp. and Campylobacter spp. was not rapid and simple as indicated by the authors. In contrary, it is quite time-consuming and tedious since it still depend on pre-enrichment process.

6. PLOS authors have the option to publish the peer review history of their article (what does this mean?). If published, this will include your full peer review and any attached files.

Reviewer #1: **Yes: **Jianchang Wang

Reviewer #2: No

---

## [Author Response · Author response to Decision Letter 0]

28 Apr 2021

Response to the reviewer’ comments

Page number indicated below refers to that of the “Revised Manuscript with Track Changes” unless stated. 

The modified text was marked in Track changes. 

Journal Requirements#:

1: Please ensure that your manuscript meets PLOS ONE's style requirements, including those for file naming. The PLOS ONE style templates can be found at

Response: The revised manuscript meets PLOS ONE's style requirements.

2: PLOS ONE now requires that authors provide the original uncropped and unadjusted images underlying all blot or gel results reported in a submission’s figures or Supporting Information files. This policy and the journal’s other requirements for blot/gel reporting and figure preparation are described in detail at https://journals.plos.org/plosone/s/figures#loc-blot-and-gel-reporting-requirements and https://journals.plos.org/plosone/s/figures#loc-preparing-figures-from-image-files. When you submit your revised manuscript, please ensure that your figures adhere fully to these guidelines and provide the original underlying images for all blot or gel data reported in your submission. See the following link for instructions on providing the original image data: https://journals.plos.org/plosone/s/figures#loc-original-images-for-blots-and-gels.

Response: Original images for all gel data reported in the revised manuscript are provided in Supporting Information file (PDF file).

Reviewer #1: General comments:

The manuscript described a d-LAMP-LFB assay for rapid, sensitive and simultaneous detection of Campylobacter spp. and Salmonella spp.in the chicken samples in one assay. In my opinion, the assay is technically sound, and the experimental data support the conclusion. There is only one minor question needed to emphasize.

Response: We thank the reviewer#1 for careful reading of our manuscript. We are grateful for your consideration of this manuscript. 

Reviewer #1: Specific comments:

Introduction:

1. Line 66; previous version: You mentioned that the real-time PCR kits for food-borne pathogens are costly. According to my knowledge, the real-time PCR is less expensive than the LAMP, especially the LAMP combined with lateral flow biosensor. 

Response: We appreciate your comment. Till date, real-time PCR kits are commercially available for detecting foodborne pathogens (1), and the cost of kits is slightly less expensive than that of developed LAMP-LFB. However, most real-time detection kits can detect only a single pathogen either Campylobacter or Salmonella species. The time and reagent cost for independently detecting these pathogens were increased. Meanwhile, d-LAMP-LFB in this study can detect both pathogens simultaneously, which was more rapid, time-saving and effort without compromising the results. The reagent cost for d-LAMP-LFB assay was reduced and found to be cheaper than that of real-time PCR kits. In addition, real-time PCR kits required real-time PCR machine, which is highly sophisticated instrument to operate. Real-time PCR technique can be used in well-equipped laboratories that is unsuitable for use in field, point-of-care and resource-limited settings (2). Meanwhile, the LAMP technique is based on auto-cycling strand displacement DNA synthesis carried out by Bst DNA polymerase at a single temperature in the range of 60-65°C to amplified nucleic acid in a single step, eliminating the use of a costly specialized instrument. Only heating block or simple water baths, which are common and cost-effective reaction equipment can be used (1, 3, 4). The advantage of low-cost and easy operation of LAMP could be adapted for use in a lack of infrastructure to support PCR facilities. In conclusion, the ability to simultaneously analyze multiple targets in a single tube can translate to increased time and cost-efficiency, yielding more data from each reaction and utilization of fewer reagents. This information is included in our revised manuscript on Page 3, Line 66-67.

Reviewer #2: General comments: 

1. The authors indicated that the d-LAMP is rapid and simple for simultaneous detection of Salmonella spp. and Campylobacter spp. However, this technique must rely on pre-enrichment which is the bottle neck of the overall process and make it not rapid and simple as claimed by the authors. Though the pre-enrichment has the major advantage in discrimination between dead and alive pathogenic bacteria, but it is tedious and time- consuming. Hence, the overall process starting from sample treatment could take approximately 24 hours or more which is not rapid and simple. As such, line 398-391 should be deleted or re-discussed since the d-LAMP combined with pre-enrichment could not be applied for field investigation.

Response: We appreciate your comment which helped improving the quality of the paper. We answer the reviewer’s concerns as:

a) This technique must rely on pre-enrichment. 

Response: To date, several molecular detection methods such as PCR, qPCR and LAMP assays have been developed for detection of pathogens from foods (1, 3). However, most are unable to detect positive samples without pre-enrichment step (5, 6). The sufficient limit of detection (LOD) of the test is important to reliably detect pathogens in foods according to the legal requirements, to reduce the chance of false negatives due to low numbers of Campylobacter and Salmonella cells in samples with high background microflora (7). Thus, an enrichment step is still necessary prior to those molecular assays (8-13). An enrichment step serves purposes as: (i) to increase bacterial cell numbers into the detection range, (ii) to discriminate between live and dead cells and (iii) to dilute interfering substances such as organic and inorganic chemicals, polysaccharides present in the sample matrix (6, 14). This information is included in our revised manuscript.

b) Overall process starting from sample treatment could take approximately 24 hours or more which is not rapid and simple.

Response: We agree with the reviewer’s comment. Our developed d-LAMP-LFB assay coupled with enrichment culture procedure could take approximately 24 hours or more. However, our d-LAMP-LFB assay provides a result as soon as next day after sample receipt within two 8-h working shifts (including 24 h of enrichment, DNA extraction and d-LAMP-LFB detection) and requires less labor and equipment. Unlike gold standard culture methods that require over 5-7 workdays for sampling, culture, enumeration, and biochemical confirmation of the presumptive bacterial colonies according to the ISO10272:2006 (Campylobacter spp.) and ISO6579:2002 (Salmonella spp.) guidelines (15, 16) are time consuming, required various instruments and labor intensive. The advantage of d-LAMP-LFB can also detect both pathogens simultaneously, which offers a more rapid detection compared to daily procedures on each standard culture method. Thus, developed d-LAMP-LFB in this study could serve as an alternative to the gold standard culture method. This information is included in our revised manuscript.

c) Line 389-391; previous version; should be deleted or re-discussed since the d-LAMP combined with pre-enrichment could not be applied for field investigation.

Response: Thank you for this remark. We apologize for such confusion. The phrase “field food samples” means “real food samples at the retail markets” to describe that our developed d-LAMP-LFB assay offers a valuable detection tool for its field applicability by detecting of both Campylobacter and Salmonella present in real food samples. As the reviewer’s suggestion, the word “field” has been removed on Page 18, Line 392 to avoid confusion.

2. One of the major food-borne pathogenic bacteria is Listeria spp. which is often found in food products. Hence, d-LAMP of Salmonella spp. and Campylobacter spp. may not be sufficient for surveillance test.

Response: Thank you for your critical reading throughout our manuscript. We agreed with the reviewer’s comments that Listeria spp. is one cause of the seriously severe foodborne diseases and it is a significant public health concern. However, according to the World Health Organization (WHO), Campylobacter and Salmonella are among the key global causes of diarrheal diseases (others including Norovirus and E. coli) (17). Both pathogens are highly prevalent in chicken meat, which is considered as the main food vehicle of infection for human (18-20). The control of these pathogens in chicken meat is also essential for avoiding sanitary barriers and reducing contamination, which is very useful for the exporter of this product (21). Thus, our d-LAMP-LFB assay developed in this work could be used for the simultaneous detection of Campylobacter and Salmonella with applications in food products. 

3. The d-LAMP condition is arbitrary for detection of both Salmonella spp. and Campylobacter spp.as well as the selected gene for primer design. Hence, this may affect the sensitivity or limit of detection of each bacterial test. Generally, the LOD of d-LAMP is less sensitive than a single target detection. 

Response: We appreciate your comment. So far, the normal molecular detection technology coupled with enrichment procedure was limited to detect only one target template either Campylobacter or Salmonella spp. in food (22, 23), as it is challenging to detect both pathogens simultaneously, due to the fact that their growth requirements on different selective broths as well as slower growth of Campylobacter under microaerobic conditions (24). In this study, we successfully detected both pathogens in only a single reaction. We found that s- and d-LAMP-LFB results showed that there was no obvious difference in the detection limit for Salmonella. However, the limit of detection of d-LAMP was found to be less sensitive by ten-time for Campylobacter. Even though, the LOD of d-LAMP less sensitive than s-LAMP in this study, however, d-LAMP sensitivity was comparable to previous publications (22, 23, 25, 26), and commercially available real-time PCR kits such as BAX® Campylobacter coli/jejuni/lari assay (DuPont Qualicon), BAX® Salmonella assay (Hygiena), BIOTECON foodproof® Campylobacter Detection Kit (Biotecon Diagnostics) and BIOTECON foodproof® Salmonella Detection Kit (Biotecon Diagnostics). Thus, our developed d-LAMP-LFB assay has the potential to produce considerable savings of time and found to be superior to other previous studies for detecting of both pathogens simultaneously. 

Are there any reasons that the authors select 16S rRNA and invB for primer and probe design? What is/are the feature (s) or advantage (s) of these genes over the others?

Response: As commented by the reviewer, according to food safety of international criteria, regulation standards have strict guidelines, stating that the analyzed sample must have no Campylobacter and Salmonella contamination (27-29). Thus, the genetic marker used for designing of LAMP primer must have conserve regions for all strains of either Campylobacter or Salmonella. In this study, our d-LAMP-LFB assay was developed for simultaneous detection of Campylobacter spp. and Salmonella spp., by targeting 16S rRNA (GenBank accession no. AL111168.1) and invB (GenBank accession no. CP009102.1) genes, respectively. The 16S rRNA gene, coding 16S ribosomal RNA, was highly conserved among Campylobacter, and it was present in the chromosomes of all Campylobacter species. It was the most frequently targeted gene for designing primers (30). Meanwhile, the Salmonella invasion protein B (invB), has been shown to be virulence factors of Salmonella (31), and this gene has been demonstrated to exist and unique within Salmonella strains but not in non-Salmonella (32). This evidence indicates that 16S rRNA and invB genes could serve as the reliable and accurate target genes for detection of all Campylobacter spp. and Salmonella spp., respectively. In addition, the specificity of our d-LAMP-LFB assay was successful investigated, and the positive results were generated in the assay of Campylobacter and Salmonella strains but not for non-Campylobacter and non-Salmonella. In conclusion, the advantage by using these target genes could be a valuable tool for detecting both target pathogens simultaneously in a variety of samples. We also added more information of these target genes in the methodology part on Page 6, Line 122-123.

4. The author should indicate the melting temperature of each primer in Table 1.

Response: As suggested by the reviewer, LAMP primer designing by Primer Explorer V5 software program (http://primerexplorer.jp/lampv5e/index.html) provided Tm value of four LAMP primers as shown in Fig 1 below. However, the forward inner primer (FIP), which consist of the F2 and F1c primers and backward inner primer (BIP), which consists of B2 and B1c primers were linked with a poly T linker “TTTT” between F1c - F2 and B1c - B2. The actual Tm value of the FIP and BIP primers were not estimated by a default setting. In addition, all of scientific articles did not include Tm value of LAMP primer in the article. Thus, we think that the melting temperature (Tm) of each primer was not necessary for the experimental design. Therefore, this information was omitted in Table 1. 

Fig 1. The detail of primer information window generated by Primer Explorer V5 software program designed on the target sequence of 16S rRNA (upper) and invB genes (lower).

<<please see the figure in the "Response to reviewer" file>>

Reviewer #2: Specific comments:

Discussion:

1. Line 415-419; previous version: are repeated to each other. This should be re-written. In my opinion detection without pre-enrichment is much more challenge.

Response: Thank you for your critical reading throughout our manuscript. We have rewritten the discussion part according to the reviewer’s concerns on Page 19, Line 417-421 The point that was concerned by the reviewer is that “detection without pre-enrichment is much more challenge”. The control of these two pathogens is essential for food safety. Detection of both organisms cannot be confirmed by the molecular techniques without an enrichment procedure (33). Thus, an enrichment process should be included in our developed LAMP-LFB assay in this study.

2. Line 422-425; previous version: The authors should point out that the less LOD of Campylobacter spp. is the limitation of d-LAMP when compared to the single detection. Hence, the primers should be re-designed or changed the gene target,

Response: We appreciate your comment. 16S rRNA and invB genes for designing LAMP primers in this study were examined as 100% specificity for Campylobacter and Salmonella, respectively. The positive results were generated in the assay of only Campylobacter and Salmonella strains, but not for non-Campylobacter and non-Salmonella. We think that these target genes are already suitable for detection of both pathogens. However, we agreed with the reviewer’s comment that successful amplification of the d-LAMP relies on the designed primers. The primer sets should have similar melting temperature (Tm) and genomic GC content to produce a successful d-LAMP assay. In this study, two set of LAMP primers for Campylobacter and Salmonella were critically examined, which showed similar Tm and GC content. However, amplification efficiency of 16S rRNA Campylobacter might be lower than that of invB Salmonella. The interaction may occur between the multiple primer set in d-LAMP that results in primer dimers or secondary structures. Thus, the condition of d-LAMP is necessary to be further optimized by adding some chemicals such as methyl sulfoxide (DMSO) could elevate amplification efficiencies by inhibiting secondary structures in the DNA primers (13, 34). We have therefore added information in the discussion part on Page 20, Line 428-432. 

3. Line 445-451; previous version: the discussion of two false positive samples are conflict to the purposes of this study. Since the samples were pre-enriched prior to d-LAMP, the contamination of dead cells is not quite possible. The second reason as the author discussed that d-LAMP-LFD could detect viable but non-culturable in XLD agar indicated that the specificity of the test is questionable. So, this is another the limitation of the test.

Response: Thanks for your careful reading of our manuscript and for giving such constructive comments which helped improving the quality of the paper. The first comment by the reviewer is that “the contamination of dead cells is not quite possible”. 

Response: Molecular techniques such as PCR, real-time PCR and LAMP were nucleic acid amplification-based techniques, which were considered with high sensitivity (1, 3). These techniques possibly detected both viable and non-viable bacteria contained in the samples (35). The high concentration of non-viable cells existence in the initial samples can be detected by those molecular techniques (9), even though food samples were enriched to dilute the number of non-viable cell and increase the ability to detect viable cell prior testing by those molecular assays (9, 35-37). Thus, this is the limitations of those nucleic acid based assays in the detection of pathogens from foods, especially in extremely sensitive detection assay like LAMP (4). Thus, an interpretation of test results by those molecular assays must be carefully considered. The addition of DNase prior DNA extraction step (38) may resolve the probability of false positives. In addition, increasing of the time for an enrichment culture step for standard culture method can be excluded the truly false-positive LAMP results for ensuring that these samples were both truly positive for LAMP and standard culture method. 

The second comment by the reviewer is that “d-LAMP-LFD could detect viable but non-culturable in XLD agar indicated that the specificity of the test is questionable. So, this is another the limitation of the test”.

Response: The viable but non-culturable bacterial cells presented in foods are also detected by molecular techniques (35). It can possibly cause false positive LAMP results by detecting of sub-lethally damaged bacterial cells (still viable) in the samples (9), which may be non-culturable on the XLD agar or under the culture conditions used in this study. This sample cannot thus be detected by the culture-based method, and it could be cultured in other selective agar of choice like Hektoen Enteric agar (HE) or Brilliance Salmonella agar (BE) according to standardized microbiological procedures (16, 39) as shown in the discussion part as well as xylose–lysine–tergitol (Niaproof) 4 agar (XLT4) and Miller-Mallinson agar (MM), which can enhance the specificity and sensitivity of Salmonella detection and the reduction of competing Enterobacteriaceae other than Salmonella. XLT4 also detected more Salmonella-positive samples of poultry origin by increasing recovery rates than did HE and XLD plates (40). These evidences indicated that using other selective agar is advantage over commonly used (XLD) recommended by the standard method for identification of Salmonella (16, 39), which might cause false positive LAMP testing. In conclusion, it is unclear whether these are truly false-positive results. The discrepancies between LAMP-LFB and culture results may also be attributed to differential-selective agar and culture conditions for isolating certain Salmonella serotypes (9). Test results must be considered for interpretation. 

4. Line 459-463; previous version: The authors mentioned the improvement of sensitivity of Campylobacter spp. using d-LAMP-LFD assay by extending incubation time or change culture medium. This confirmed that the detection of Campylobacter spp. was not rapid and simple as indicated by the authors. 

Response: We appreciate your comment. We proposed the take home massage that the one false negative reported in this study was probably due to Campylobacter cell density, being below the minimum detection limit of d-LAMP-LFB. An enrichment could circumvent this problem by bringing Campylobacter cell numbers into the detection range of d-LAMP-LFB assay. However, Campylobacter is fastidious and slow-growing organisms in enrich culture. They required enrichment time for at least approximately 48 h under microaerobic conditions (21, 41), while our enrichment time for 24 h is enough for detecting Campylobacter, which was comparable to other previous studies and commercially available real-time PCR kits. We have therefore rewritten this point in the discussion part on Page 21, Line 466-468 and Page 22, line 469.

5. Line 480-485; previous version: The discussion is unacceptable due to the d-LAMP LFD assay has some limitations as followed.

a. The assay required pre-enrichment step which is time-consuming and tedious.

Response: As commented by the Reviewer, our developed d-LAMP-LFB assay coupled with enrichment procedure can be used for detecting both Campylobacter and Salmonella in foods simultaneously, which provides a test result within two 8-h working shifts, more time-efficient and labor saving than that of the gold standard culture-based method. Even though the molecular technique is a highly sensitive detection method, but it is not of value if applied directly for the detection of pathogens in foods (6). An enrichment step is still required for detecting pathogens in foods not only enhance the cell number prior to testing by molecular detection method but also inhibits the growth of background flora and dilutes inhibitor present in different food matrices, which can interfere amplification reaction (8). Without requiring enrichment step prior molecular testing, false-negative testing may occur, which increases the risk of foodborne illness in human due to contaminated food. As well, false-negative test results could be of critical importance to food safety regulation whose samples are compromised when positive sample is misinterpreted as negative. Thus, the inclusion of a minimum enrichment time for at least 24 h is necessary in a detection protocol in this study.

 b. The LOD of the assay is less sensitive than a single detection especially for Campylobacter spp.

Response: As commented by the Reviewer, the LOD for Campylobacter of our developed d-LAMP-LFB assay is less sensitive than that of a single detection. However, the detection range of LOD of our assay is comparable to a previous study (22) and commercially available real-time PCR rapid detection kits. In addition, d-LAMP-LFB assay here can potentially facilitate simultaneous detection and correctly distinguish Campylobacter and Salmonella in only a single reaction, which is considered more rapid and convenient than a single testing. Therefore, developed d-LAMP-LFB assay could be an alternative method to rapidly detect both pathogens in foods.

c. The specificity of the assay is questionable since the viable non-culturable in XLD agar could interfere the results.

Response: Thank you for pointing this out. Two false positive LAMP results were interpreted for Salmonella detection. We have already discussed that these false positive results may probably due to detect dead cells and/or viable-non culturable cells in the discussion part. We also drew the schematic representation of the standard culture method compared with LAMP-LFB assay in Fig 2 below. We think that our developed d-LAMP-LFB detected false positive results due to the existence of sub-lethally damaged Salmonella cells (still viable) in the samples, which may be non-culturable (9) on the medium (XLD) used in this study. It may by cultured on differential-selective agar and culture conditions for isolating certain Salmonella (9) serotypes such as Hektoen Enteric agar (HE), Brilliance Salmonella agar (BE), xylose–lysine–tergitol (Niaproof) 4 agar (XLT4) and Miller-Mallinson agar (MM). Two false positive LAMP results in this study may not be considered as a truly-false positive result. Thus, the result interpretation could explain the limited detection specificity of molecular method, especially when false positive result was present in testing

6. In conclusion of my comment, this confirmed that the d-LAMP-LFD for detection of Salmonella spp. and Campylobacter spp. was not rapid and simple as indicated by the authors. In contrary, it is quite time-consuming and tedious since it still depends on pre-enrichment process.

Response: Thank you very much for your comment. Because of the increasing importance of Salmonella spp. and Campylobacter spp., rapid and simple methods for their detection are required. Even though the culture-based method is still used as the gold standard for detection of these pathogens, the routine procedures required multiple subculture steps, isolation, confirmation and species identification, which required about 7 days for a conclusive negative result and 7 to 14 days for confirmation of a positive finding (15, 16). They have their own limitation including time-consuming and labor intensive (1, 3). Our developed d-LAMP-LFB assay in contrast allow rapid detection of both pathogens simultaneously within two 8-h working shifts, more time-efficient and labor saving. An additional of enrichment step in procedure enables molecular detection assays to efficiently and accurately detected low numbers of pathogens in food samples, so this step is still essential for nucleic acid based assays to reduce the chance of false negatives result (7-12). In addition, our d-LAMP-LFB assay is also a cost-effective and easy to perform under isothermal condition using just only simple water bath, eliminating the use of highly sophisticated instrument. Thus, our developed d-LAMP-LFB assay serves as an alternative tool to standard culture method with the advantage of less time consuming in detecting both pathogens simultaneously. 

References

1. Law JW, Ab Mutalib NS, Chan KG, Lee LH. Rapid methods for the detection of foodborne bacterial pathogens: principles, applications, advantages and limitations. Frontiers in microbiology. 2014;5:770.

2. Dhama K, Karthik K, Chakraborty S, Tiwari R, Kapoor S, Kumar A, et al. Loop-mediated isothermal amplification of DNA (LAMP): a new diagnostic tool lights the world of diagnosis of animal and human pathogens: a review. Pakistan journal of biological sciences : PJBS. 2014;17(2):151-66.

3. Zhao X, Lin CW, Wang J, Oh DH. Advances in rapid detection methods for foodborne pathogens. Journal of microbiology and biotechnology. 2014;24(3):297-312.

4. Wong Y-P, Othman S, Lau Y-L, Radu S, Chee H-Y. Loop-mediated isothermal amplification (LAMP): a versatile technique for detection of micro-organisms. J Appl Microbiol. 2018;124(3):626-43.

5. Myint MS, Johnson YJ, Tablante NL, Heckert RA. The effect of pre-enrichment protocol on the sensitivity and specificity of PCR for detection of naturally contaminated Salmonella in raw poultry compared to conventional culture. Food Microbiol. 2006;23(6):599-604.

6. Kaneko I, Miyamoto K, Mimura K, Yumine N, Utsunomiya H, Akimoto S, et al. Detection of enterotoxigenic Clostridium perfringens in meat samples by using molecular methods. Appl Environ Microbiol. 2011;77(21):7526-32.

7. Lim HS, Zheng Q, Miks-Krajnik M, Turner M, Yuk HG. Evaluation of commercial kit based on loop-mediated isothermal amplification for rapid detection of low levels of uninjured and injured Salmonella on duck meat, bean sprouts, and fishballs in Singapore. Journal of food protection. 2015;78(6):1203-7.

8. Cheng CM, Lin W, Van KT, Phan L, Tran NN, Farmer D. Rapid detection of Salmonella in foods using real-time PCR. Journal of food protection. 2008;71(12):2436-41.

9. Hong Y, Berrang ME, Liu T, Hofacre CL, Sanchez S, Wang L, et al. Rapid detection of Campylobacter coli, C. jejuni, and Salmonella enterica on poultry carcasses by using PCR-enzyme-linked immunosorbent assay. Appl Environ Microbiol. 2003;69(6):3492-9.

10. Mester P, Wagner M, Rossmanith P. Molecular Enrichment for Qualitative Molecular Pathogen Detection in Food. Food Analytical Methods. 2018;11(5):1251-6.

11. Yamazaki W, Taguchi M, Kawai T, Kawatsu K, Sakata J, Inoue K, et al. Comparison of loop-mediated isothermal amplification assay and conventional culture methods for detection of Campylobacter jejuni and Campylobacter coli in naturally contaminated chicken meat samples. Appl Environ Microbiol. 2009;75(6):1597-603.

12. Cremonesi P, Pisani LF, Lecchi C, Ceciliani F, Martino P, Bonastre AS, et al. Development of 23 individual TaqMan® real-time PCR assays for identifying common foodborne pathogens using a single set of amplification conditions. Food Microbiol. 2014;43:35-40.

13. Yang Q, Domesle KJ, Ge B. Loop-mediated isothermal amplification for Salmonella detection in food and feed: current applications andfFuture directions. Foodborne pathogens and disease. 2018;15(6):309-31.

14. Thung T, Lee E, Wai G, Pui C, Kuan C, Premarathne J, et al. A review of culture-dependent and molecular methods for detection of Salmonella in food safety. Food Research 2019;3(6):622-7.

15. ISO10272:2006(E). Microbiology of food and animal feeding stuffs-horizontal method for the detection and enumeration of Campylobacter spp., International Standard Organization.

16. ISO6579:2002(E). Microbiology of food and animal feeding stuffs-horizontal method for the detection of Salmonella spp., International Standard Organization.

17. WHO. Diarrhoeal diseases: WHO estimates of the global burden of foodborne diseases. 2015. Available from: https://www.who.int/foodsafety/areas_work/foodborne-diseases/infographics_combined_en.pdf.

18. WHO. Salmonella and Campylobacter in chicken meat, microbiological risk assessment series 19, FAO/WHO expert meeting report. [revised 2009; cited 2020 Aug 1]. Available from: https://www.who.int/foodsafety/publications/mra19/en/.

19. Thames HT, Theradiyil Sukumaran A. A review of Salmonella and Campylobacter in broiler meat: emerging challenges and food safety measures. Foods (Basel, Switzerland). 2020;9(6):776.

20. Rukambile E, Sintchenko V, Muscatello G, Kock R, Alders R. Infection, colonization and shedding of Campylobacter and Salmonella in animals and their contribution to human disease: a review. Zoonoses and public health. 2019;66(6):562-78.

21. Alves J, Marques VV, Pereira LFP, Hirooka EY, De Oliveira TCRM. Multiplex PCR for the detection of Campylobacter spp. and Salmonella spp. in chicken meat. Journal of Food Safety. 2012;32(3):345-50.

22. Thongphueak D, Chansiri K, Sriyapai T, Areekit S, Santiwatanakul S, Wangroongsarb P. Development of the rapid test kit for the identification of Campylobacter spp. based on loop-mediated isothermal amplification (LAMP) in combination with a lateral flow dipstick (LFD) and gold nano-DNA probe (AuNPs). Science & Technology Asia. 2019;24(1):63-71.

23. Liu Z, Zhang Q, Yang NN, Xu MG, Xu JF, Jing ML, et al. Rapid and sensitive detection of Salmonella in chickens using loop-mediated isothermal amplification combined with a lateral flow dipstick. Journal of microbiology and biotechnology. 2019;29(3):454-64.

24. Wolffs PF, Glencross K, Norling B, Griffiths MW. Simultaneous quantification of pathogenic Campylobacter and Salmonella in chicken rinse fluid by a flotation and real-time multiplex PCR procedure. International journal of food microbiology. 2007;117(1):50-4.

25. Yu J, Xing J, Zhan X, Yang Z, Qi J, Wei Y, et al. Improvement of loop-mediated isothermal amplification combined with chromatographic flow dipstick assay for Salmonella in food samples. Food Analytical Methods. 2020;13(7):1398-408.

26. Mei X, Zhai X, Lei C, Ye X, Kang Z, Wu X, et al. Development and application of a visual loop-mediated isothermal amplification combined with lateral flow dipstick (LAMP-LFD) method for rapid detection of Salmonella strains in food samples. Food Control. 2019;104:9-19.

27. FSANZ/Australia New Zealand. Food Standards Australia New Zealand, Compendium of microbiological criteria for food. [revised 2018 Sep ; cited 2020 Aug 1]. Available from: https://www.foodstandards.gov.au/publications/Documents/Compedium%20of%20Microbiological%20Criteria/Compendium_revised-jan-2018.pdf.

28. EC. Commission Regulation (EC) No 2073/ 2005 of 15 November 2005 on microbiological criteria for foodstuffs. [revised 2005 Nov 15 ; cited 2020 Aug 1]. Available from: https://eur-lex.europa.eu/legal-content/EN/TXT/PDF/?uri=CELEX:02005R2073-20140601&from=EN.

29. DLD/Thailand. Department of Livestock Development,Thailand. Microbiological guideline for chilled / frozen meat & poultry meat. [revised 2008; cited 2020 Aug 1]. Available from: http://qcontrol.dld.go.th/images/law/regulation/MicrobiologicalSTDforLivestockProducts.PDF.

30. Ricke SC, Feye KM, Chaney WE, Shi Z, Pavlidis H, Yang Y. Developments in rapid detection methods for the detection of foodborne Campylobacter in the United States. Frontiers in microbiology. 2018;9:3280.

31. Sirsat SA, Burkholder KM, Muthaiyan A, Dowd SE, Bhunia AK, Ricke SC. Effect of sublethal heat stress on Salmonella Typhimurium virulence. J Appl Microbiol. 2011;110(3):813-22.

32. Porter SB, Curtiss R, 3rd. Effect of inv mutations on Salmonella virulence and colonization in 1-day-old White Leghorn chicks. Avian diseases. 1997;41(1):45-57.

33. Alves J, Hirooka EY, de Oliveira TCRMJL-FS, Technology. Development of a multiplex real-time PCR assay with an internal amplification control for the detection of Campylobacter spp. and Salmonella spp. in chicken meat. LWT Food Sci Technol. 2016;72:175-81.

34. Shao Y, Zhu S, Jin C, Chen F. Development of multiplex loop-mediated isothermal amplification-RFLP (mLAMP-RFLP) to detect Salmonella spp. and Shigella spp. in milk. International journal of food microbiology. 2011;148(2):75-9.

35. Saiyudthong S, Phusri K, Buates S. Rapid detection of Campylobacter jejuni, Campylobacter coli, and Campylobacter lari in fresh chicken meat and by-products in Bangkok, Thailand, using modified multiplex PCR. Journal of food protection. 2015;78(7):1363-9.

36. Sridapan T, Tangkawsakul W, Janvilisri T, Kiatpathomchai W, Dangtip S, Ngamwongsatit N, et al. Rapid detection of Clostridium perfringens in food by loop-mediated isothermal amplification combined with a lateral flow biosensor. PloS one. 2021;16(1):e0245144.

37. Siala M, Barbana A, Smaoui S, Hachicha S, Marouane C, Kammoun S, et al. Screening and detecting Salmonella in different food matrices in southern Tunisia using a combined enrichment/real-time PCR method: correlation with conventional culture method. Frontiers in microbiology. 2017;8:2416.

38. Priya GB, Agrawal RK, Milton AAP, Mishra M, Mendiratta SK, Agarwal RK, et al. Development and evaluation of isothermal amplification assay for the rapid and sensitive detection of Clostridium perfringens from chevon. Anaerobe. 2018;54:178-87.

39. FDA-BAM. Bacteriological analytical manual chapter 5: Salmonella. Food & Drug Administration, Bacteriological analytical manual, U.S. Department of health and human services, Public health service, Washington DC. [revised 2020 Jul ; cited 2020 Aug 1]. Available from: https://www.fda.gov/food/laboratory-methods-food/bam-chapter-5-salmonella.

40. Mallinson ET, Miller RG, de Rezende CE, Ferris KE, deGraft-Hanson J, Joseph SW. Improved plating media for the detection of Salmonella species with typical and atypical hydrogen sulfide production. Journal of veterinary diagnostic investigation : official publication of the American Association of Veterinary Laboratory Diagnosticians, Inc. 2000;12(1):83-7.

41. Sails AD, Bolton FJ, Fox AJ, Wareing DR, Greenway DL. Detection of Campylobacter jejuni and Campylobacter coli in environmental waters by PCR enzyme-linked immunosorbent assay. Appl Environ Microbiol. 2002;68(3):1319-24.

---

## [Decision Letter · Decision Letter 1]

11 May 2021

PONE-D-21-06858R1

Rapid and simultaneous detection of Campylobacter spp. and Salmonella spp. in chicken samples by duplex loop-mediated isothermal amplification coupled with a lateral flow biosensor assay

PLOS ONE

Dear Dr. Chankhamhaengdecha,

Thank you for submitting your manuscript to PLOS ONE. After careful consideration, we feel that it has merit but does not fully meet PLOS ONE’s publication criteria as it currently stands. Therefore, we invite you to submit a revised version of the manuscript that addresses the points raised during the review process.

Minor revisions needed including the limit of detection claimed for the method described. 

We look forward to receiving your revised manuscript.

Kind regards,

Arun K. Bhunia, Ph.D.

Academic Editor

PLOS ONE

Journal Requirements:

Additional Editor Comments (if provided):

Authors are requested to address the minor comments raised during second review including LOD.

Reviewers' comments:

Reviewer's Responses to Questions

**Comments to the Author**

1. If the authors have adequately addressed your comments raised in a previous round of review and you feel that this manuscript is now acceptable for publication, you may indicate that here to bypass the “Comments to the Author” section, enter your conflict of interest statement in the “Confidential to Editor” section, and submit your "Accept" recommendation.

Reviewer #1: All comments have been addressed

Reviewer #3: All comments have been addressed

2. Is the manuscript technically sound, and do the data support the conclusions?

Reviewer #1: Yes

Reviewer #3: Yes

3. Has the statistical analysis been performed appropriately and rigorously? 

Reviewer #1: Yes

Reviewer #3: N/A

4. Have the authors made all data underlying the findings in their manuscript fully available?

Reviewer #1: Yes

Reviewer #3: Yes

5. Is the manuscript presented in an intelligible fashion and written in standard English?

Reviewer #1: Yes

Reviewer #3: Yes

6. Review Comments to the Author

Reviewer #1: (No Response)

Reviewer #3: The manuscript PONE-D-21-06858R1, reports the development and application of the d-LAMP-LFB for the multiplex detection of two foodborne pathogens simultaneously. The article is well written and presents a great deal of work for the assessment of the important possibility to detect the presence of several food-borne pathogens in only one reaction tube and test. The authors claimed that is the first multiplex LAMP-LFB for detection of Campylobacter spp and Salmonella spp.

Major comments

There is a constant misunderstand between limit of detection (LOD) of an assay and the initial spiking/initial inoculum of bacterial cells. LOD is defined as the minimum detectable amount/concentration of the target (bacteria cells or DNA) on the sample prior to application in the tests. The initial inoculum is the initial bacterial concentration on sample before the enrichment in a selective medium. Authors should not interpret that low levels of initial inoculation used for the enrichment step are the LOD of the test. Therefore, we recommend that authors revise and explain better when they discuss and compare the LOD.

Please, see the specific comments below.

Line 228 – Please note that “the last dilution in each spiked sample that test positive” is not the detection limit (LOD) of the test but the inoculation level. This is a major concern with many other papers discussing the LOD. Anytime there is enrichment, the initial spiking level becomes irrelevant for determining the LOD of the test. What is more important is how many cells are going into the sample used in your LAMP assay. Nevertheless, it is important to mention the initial spiking level but it is wrong to use it as the LOD.

Lines 352-355 – Authors are reporting in this paragraph the initial spiking level that is the sample analytical LOD but not the LOD of the LAMP-LFB.

Figure 5 – Please, report as initial spiking level or sample analytical LOD.

Lines 432- 445 – I don’t think that is the appropriate manner to discuss the results obtained since the authors did not determined how many CFU/ml they had in their enriched sample before DNA extraction and application on the LAMP-LFB. In addition, by claiming 0.04 CFU/g the authors are considering that they initiated the experiments by inoculating only one bacterium. True comparison of tests should also evaluate whole assay time. As the majority of tests and regulation standards seek to detect a single bacterium in 25g of sample, the spiking level is irrelevant for comparison.

Minor comments

Lines 54-55 – Authors would emphasize the importance of their work by adding in the introduction few sentences with information on the main bacteria causing diarrheal diseases and the recommended control rules and not just mention the two studied bacteria and being so vague as mentioning “so the regulation standards have strict guidelines”.

Line 60 – The article reports about a test to detect the presence of bacteria therefore it is not a diagnostic test.

Line 65 – The authors can increase the value of their work by mentioning the other disadvantages of PCR in order to emphasize the advantage to use d-LAMP-LFB.

Line 103 – Please, inform the source of the BHI broth applied.

7. PLOS authors have the option to publish the peer review history of their article (what does this mean?). If published, this will include your full peer review and any attached files.

Reviewer #1: **Yes: **Jianchang Wang

Reviewer #3: **Yes: **Samira Bührer-Sékula

---

## [Author Response · Author response to Decision Letter 1]

11 Jun 2021

Page numbers and lines mentioned refers to that of the main text of the revised manuscript with inline markup version.

Journal Requirements#:

1: Please review your reference list to ensure that it is complete and correct. If you have cited papers that have been retracted, please include the rationale for doing so in the manuscript text, or remove these references and replace them with relevant current references. Any changes to the reference list should be mentioned in the rebuttal letter that accompanies your revised manuscript. If you need to cite a retracted article, indicate the article’s retracted status in the References list and also include a citation and full reference for the retraction notice.

Response: Thanks for your careful reading of our manuscript and for giving such constructive comments which helped improving the quality of the paper. The reference number 3, 6, 7, 8, 26, 27, 33, 36, 39, and 44 have been reviewed and replaced with relevant current references to ensure that the citation is complete and accurate. 

Reviewer #3: General comments:

The manuscript PONE-D-21-06858R1, reports the development and application of the d-LAMP-LFB for the multiplex detection of two foodborne pathogens simultaneously. The article is well written and presents a great deal of work for the assessment of the important possibility to detect the presence of several food-borne pathogens in only one reaction tube and test. The authors claimed that is the first multiplex LAMP-LFB for detection of Campylobacter spp. and Salmonella spp.

Response: We thank the reviewer#3 for careful reading of our manuscript. We are grateful for your consideration of this manuscript. 

Reviewer #3: Major comments:

There is a constant misunderstand between limit of detection (LOD) of an assay and the initial spiking/initial inoculum of bacterial cells. LOD is defined as the minimum detectable amount/concentration of the target (bacteria cells or DNA) on the sample prior to application in the tests. The initial inoculum is the initial bacterial concentration on sample before the enrichment in a selective medium. Authors should not interpret that low levels of initial inoculation used for the enrichment step are the LOD of the test. Therefore, we recommend that authors revise and explain better when they discuss and compare the LOD. 

Response: Thank you for your critical reading throughout our manuscript. We agreed with the reviewer’s comments that limit of detection (LOD) is important performance characteristics in method validation. Analytical sensitivity or LOD are used to describe the lowest concentration of an analyte that can be measured by an analytical procedure. In this study, the analytical sensitivity was used to assess the analysis of the lowest concentration of serially diluted genomic DNA templates as the “sensitivity limit”. Meanwhile, in spiking study, the last dilution in each spiked sample that tested positive was defined as “the lowest inoculated detection limit” of d-LAMP-LFB assay for visualization of the LAMP products. This term has also been used in previous publications to determine the analytical sensitivity in spiking study (Techathuvanan et al., 2010). We therefore have rewritten these terms on Page 2, Line 32; Page 6, Line 121; Page 9, Line 194;

Page 11, Line 236, 238, 239 and 242; Page 16, Line 338, 339, 340, 341, 343, 348, 349, 350, 354, 356 and 357; Page 17, Line 358, 360, 361, 362, 365 and 367; Page 18, Line 398; Page 20, Line 433, 440, 441 and 441; Page 22, Line 481 and 483.

Reference: Techathuvanan C, Draughon FA, D'Souza DH. Loop-mediated isothermal amplification (LAMP) for the rapid and sensitive detection of Salmonella Typhimurium from pork. Journal of food science. 2010;75(3):M165-72.

Reviewer #3: Specific comments:

Introduction:

1. Line 54-55; previous version: Authors would emphasize the importance of their work by adding in the introduction few sentences with information on the main bacteria causing diarrheal diseases and the recommended control rules and not just mention the two studied bacteria and being so vague as mentioning “so the regulation standards have strict guidelines”.

Response: We appreciate your comment. We revised the manuscript accordingly, please kindly see Page 3, Line 48-58.

2. Line 60; previous version: The article reports about a test to detect the presence of bacteria therefore it is not a diagnostic test.

Response: We appreciate your comment. The phrase “establishing the laboratory diagnostic of these pathogens” has been rephrased to “detecting these pathogens present in food” on Page 3, Line 64-65.

3. Line 65; previous version: The authors can increase the value of their work by mentioning the other disadvantages of PCR in order to emphasize the advantage to use d-LAMP-LFB.

Response: We appreciate your comment which helped improving the quality of the paper. To emphasize the advantage to use d-LAMP-LFB, we added more information of the advantage to use d-LAMP-LFB on Page 4, Line 87-89. 

Methodology:

1. Line 103; previous version: Please, inform the source of the BHI broth applied.

Response: Thank you for this remark. The phrase “Himedia, India” has been added on Page 5, Line 110.

2. Line 228; previous version: Please note that “the last dilution in each spiked sample that test positive” is not the detection limit (LOD) of the test but the inoculation level. This is a major concern with many other papers discussing the LOD. Anytime there is enrichment, the initial spiking level becomes irrelevant for determining the LOD of the test. What is more important is how many cells are going into the sample used in your LAMP assay. Nevertheless, it is important to mention the initial spiking level but it is wrong to use it as the LOD.

Response: We appreciate your comment. We agreed with the reviewer’s comments. The LOD used in this study was the initial spiking level or lowest inoculated detection limits. Thus, the LOD in this study has been rephrased as lowest inoculated detection limits on Page 11, Line 236.

Result:

1. Line 352-355; previous version: Authors are reporting in this paragraph the initial spiking level that is the sample analytical LOD but not the LOD of the LAMP-LFB.

Response: As commented by the reviewer, the LOD of the LAMP-LFB was rephrased as lowest inoculated detection limits on Page 16, Line 354-362.

2. Line 355; Figure 5; previous version: Please, report as initial spiking level or sample analytical LOD.

Response: As suggested by the reviewer, the LOD of the LAMP-LFB was rephrased as lowest inoculated detection limits on Page 17, Line 365 and 367.

Discussion:

1. Line 432-445; previous version: I don’t think that is the appropriate manner to discuss the results obtained since the authors did not determine how many CFU/ml they had in their enriched sample before DNA extraction and application on the LAMP-LFB. In addition, by claiming 0.04 CFU/g the authors are considering that they initiated the experiments by inoculating only one bacterium. True comparison of tests should also evaluate whole assay time. As the majority of tests and regulation standards seek to detect a single bacterium in 25 g of sample, the spiking level is irrelevant for comparison.

Response: Thank you for your critical reading throughout our manuscript. We have rewritten the discussion part on Page 20, Line 441-458, according to the reviewer’s concerns on comparing the lowest inoculated detection limits and the time for enrichment procedure of our d-LAMP-LFB assay with other previous studies. The parenthesis indicated the lowest inoculated detection limits of the original report of other previous studies. 

In addition, we apologize for such confusion at discussion part on page 21, line 465-467. This sentence is a redundancy sentence, and it has the same meaning and already describe on discussion part on page 22, line 487-499. Thus, these sentences on page 21, line 465-467 have been removed from the manuscript.

---

## [Editor Report · Decision Letter 2]

18 Jun 2021

Rapid and simultaneous detection of Campylobacter spp. and Salmonella spp. in chicken samples by duplex loop-mediated isothermal amplification coupled with a lateral flow biosensor assay

PONE-D-21-06858R2

Dear Dr. Chankhamhaengdecha,

We’re pleased to inform you that your manuscript has been judged scientifically suitable for publication and will be formally accepted for publication once it meets all outstanding technical requirements.

Kind regards,

Arun K. Bhunia, Ph.D.

Academic Editor

PLOS ONE

Additional Editor Comments (optional):

Accept
---

## [Editor Report · Acceptance letter]

23 Jun 2021

PONE-D-21-06858R2 

Rapid and simultaneous detection of *Campylobacter* spp. and Salmonella spp. in chicken samples by duplex loop-mediated isothermal amplification coupled with a lateral flow biosensor assay 

Dear Dr. Chankhamhaengdecha:

I'm pleased to inform you that your manuscript has been deemed suitable for publication in PLOS ONE. Congratulations! Your manuscript is now with our production department. 

Kind regards, 

on behalf of

Dr. Arun K. Bhunia 

Academic Editor

PLOS ONE